# Implementation of trait-based ozone plant sensitivity in the Yale Interactive terrestrial Biosphere model v1.0 to assess global vegetation damage

Yimian Ma [1, 2], Xu Yue [3*], Stephen Sitch [4*], Nadine Unger [3], Johan Uddling [5], Lina M. Mercado [4, 6], Cheng Gong [7], Zhaozhong Feng [8], Huiyi Yang [9], Hao Zhou [1, 2], Chenguang Tian [1, 2], Yang Cao [1, 2], Yadong Lei [10], Alexander W. Cheesman [4, 11], Yansen Xu [8], Maria Carolina Duran Rojas [12]

[1] Climate Change Research Center, Institute of Atmospheric Physics, Chinese Academy of Sciences, Beijing, 100029, China
[2] University of Chinese Academy of Sciences, Beijing, 100029, China
[3] Jiangsu Key Laboratory of Atmospheric Environment Monitoring and Pollution Control, Jiangsu Collaborative Innovation Center of Atmospheric Environment and Equipment Technology, School of Environmental Science and Engineering, Nanjing University of Information Science and Technology, Nanjing, 210044, China
[4] Faculty of Environment, Science and Economy, University of Exeter, Exeter, EX4 4RJ, UK
[5] Department of Biological and Environmental Sciences, University of Gothenburg, Gothenburg, P.O. Box 461, 40530, Sweden
[6] UK Centre for Ecology and Hydrology, Benson Lane, Wallingford, OX10 8BB, UK
[7] State Key Laboratory of Atmospheric Boundary Layer Physics and Atmospheric Chemistry (LAPC), Institute of Atmospheric Physics, Chinese Academy of Sciences, Beijing, 100029, China
[8] School of Applied Meteorology, Nanjing University of Information Science and Technology, Nanjing, 210044, China
[9] Livelihoods and Institutions Department, Natural Resources Institute, University of Greenwich, Kent, ME4 4TB, UK
[10] Chinese Academy of Meteorological Sciences, Beijing, 100081, China
[11] Centre for Tropical Environmental and Sustainability Science, College of Science & Engineering, James Cook University, Cairns, Queensland, 4870 Australia
[12] College of Engineering, Mathematics, and Physical Sciences, University of Exeter, Exeter, EX4 4PY, UK

*Correspondence to*: Xu Yue (yuexu@nuist.edu.cn) and Stephen Sitch (S.A.Sitch@exeter.ac.uk)


**Abstract**


A major limitation in modeling global ozone ($O_3$) vegetation damage has long been the reliance on
empirical $O_3$ sensitivity parameters derived from a limited number of species and applied at the level of
plant functional types (PFTs), which ignore the large interspecific variations within the same PFT. Here,
we present a major advance in large-scale assessments of $O_3$ plant injury by linking the trait leaf mass per
area (LMA) and plant $O_3$ sensitivity in a broad and global perspective. Application of the new approach
and a global LMA map in a dynamic global vegetation model reasonably represents the observed
interspecific responses to $O_3$ with a unified sensitivity parameter for all plant species. Simulations suggest
a contemporary global mean reduction of 4.8% in gross primary productivity by $O_3$, with a range of 1.1%-
12.6% for varied PFTs. Hotspots with damages > 10% are found in agricultural areas in the eastern U.S.,
western Europe, eastern China, and India, accompanied by moderate to high levels of surface $O_3$.
Furthermore, we simulate the distribution of plant sensitivity to $O_3$, which is highly linked with the
inherent leaf trait trade-off strategies of plants, revealing high risks for fast-growing species with low
LMA, such as crops, grasses and deciduous trees.

## 1. Introduction

Tropospheric ozone ($O_3$) has long been recognized as a hazardous pollutant for plants (Richards et al., 1958; Reich and Amundson, 1985). As a strong oxidant, $O_3$ can cause damage to leaf cells (Feng et al., 2014), impact stomata conductance (Buker et al., 2015), and reduce photosynthesis and biomass (Wittig et al., 2009). These negative impacts dampen global plant productivity (Ainsworth et al., 2012; Ainsworth et al., 2020) and crop yield (Tai et al., 2014; Emberson et al., 2018; Feng et al., 2022), altering multiple ecosystem functions and services across various spatiotemporal scales (Agathokleous et al., 2020; Feng et al., 2021). Thus, it is of crucial importance to quantify $O_3$ plant damage in global modeling and assess its coupling effects in the biosphere-atmosphere systems (Zhou et al., 2018).

To date, $O_3$ fumigation experiments have been conducted for various plant species. Accordingly, $O_3$ damage sensitivities, denoted as the Dose-Response Relationships (DRRs), were derived as the regressions between $O_3$ exposure metrics and the changes in biotic indicators (Mills et al., 2011). The widely-used $O_3$ metrics include ambient $O_3$ concentrations for AOT40 (Accumulated $O_3$ concentration above the Threshold of 40 ppbv (Fuhrer et al., 1997)), or the stomatal $O_3$ flux for $POD_y$ (Phytotoxic $O_3$ Dose above a threshold flux of y (Buker et al., 2015)). The biotic indicators include visual leaf states, photosynthetic rate, biomass, or crop yield. Normally, the DRRs were derived for typical tree/grass species in specific regions, for example, Norway spruce, birch, and beech in Europe (Buker et al., 2015) or poplar (Shang et al., 2017) and crops (Peng et al., 2019) in East Asia.

Some assessment studies used DRRs to derive contemporary $O_3$ plant damage patterns at large scales. Concentration-based DRRs were widely measured and applied on the homogenized land cover, mostly for estimating crop yield loss (Feng et al., 2022; Tai et al., 2021; Hong et al., 2020). However, such DRRs do not include information about biochemical defense and stomatal regulation. In comparison, flux-based DRRs reflect a more detailed consideration in biological processes, but are limited by the application scales in both space and time (Mills et al., 2011; Mills et al., 2018b). For example, the estimate of $POD_y$ needs a dry deposition model "$DO_3SE$" (Deposition of Ozone for Stomatal Exchange) (Clrtap, 2017) or an equivalent model to account for environmental constraints on plant stomatal uptake during the whole

growing season. Furthermore, the application of DRRs might introduce uncertainties due to the omission of complex interactions among biotic and abiotic factors at varied spatiotemporal scales.

Alternatively, more and more mechanistic schemes were developed and implemented in dynamic global vegetation models (DGVMs) to assess the joint effects of environmental factors and $O_3$ on plants. Felzer et al. (2004) considered both the damaging (through AOT40) and healing (through growth) processes related to $O_3$ effects within the framework of Terrestrial Ecosystem Model. They further estimated the reduction of 2.6%-6.8% in the net primary productivity by $O_3$ pollution in U.S. during 1980-1990. Different from Felzer et al. (2004), Sitch et al. (2007) proposed a flux-based scheme linking the instantaneous $POD_y$ with plant damage through the coupling between stomatal conductance and photosynthetic rate. Implementing this scheme into the vegetation model of YIBs, Yue and Unger (2015) predicted a range of 2%-5% reduction in global gross primary productivity (GPP) taking into account the low to high $O_3$ sensitivities for each vegetation type. Lombardozzi et al. (2015) collected hundreds of measurements and derived the decoupled responses on stomatal conductance and photosynthesis for the same $O_3$ uptake fluxes. They further implemented the separate response relationships into the Community Land Model and estimated a reduction of 8%-12% in GPP by $O_3$ at present day. Coupling these schemes with earth system models, studies have assessed interactive $O_3$ impacts on the carbon sink (Oliver et al., 2018; Yue and Unger, 2018), global warming (Sitch et al., 2007), and air pollution (Zhou et al., 2018; Gong et al., 2020; Gong et al., 2021; Zhu et al., 2022).

Although different schemes considered varied physical processes (Ollinger et al., 1997; Felzer et al., 2004; Sitch et al., 2007; Felzer et al., 2009; Lombardozzi et al., 2015; Oliver et al., 2018), they followed the same principle that different $O_3$ sensitivities should be applied for varied plant functional types (PFTs), as revealed by many measurements in the past four decades (Buker et al., 2015; Mills et al., 2018b) (Table S1). Generally, needleleaf trees, deciduous woody plants, and crop species show ascending sensitivities to $O_3$ (Reich and Amundson, 1985; Davison and Barnes, 1998; Buker et al., 2015). But the cause of such variation is not fully understood and thus has not been uniformly described in vegetation models (Massman et al., 2000; Tiwari et al., 2016). As a result, all large-scale assessments of $O_3$ vegetation

damage had to rely on a PFT-based range of sensitivity parameters derived from a limited number of plant species and attempted to envelop the range of $O_3$ impacts by assuming all species within a PFT are either "high" or "low" sensitive to $O_3$. For example, Felzer et al. (2004) defined empirical sensitivity coefficients for three major plants including deciduous trees, coniferous trees, and crops. In Sitch et al. (2007), the sensitivity coefficients were defined separately for five PFTs with high/low ranges calibrated by DRRs of typical species. These synthesized assumptions cannot resolve the intra-PFT variations in the $O_3$ sensitivity and thus may cause large uncertainties in regional to global assessments.

Recent observations revealed a uniform plant sensitivity to $O_3$ if stomatal $O_3$ flux was expressed based on a leaf mass rather than leaf area basis (Li et al., 2016; Feng et al., 2018; Li et al., 2022). The trait of leaf mass per area (LMA) is an important metric linking leaf area to mass. In a comparative study with 21 woody species (Li et al., 2016) and a meta-analysis of available experimental data (Feng et al., 2018), the DRR showed convergent $O_3$ sensitivities for conifer and broadleaf trees if the area-based stomatal uptake was converted to the mass-based flux with LMA. This is likely related to the diluting effect of thicker leaves, which normally have stronger defenses against $O_3$ in their cross-section. Nowadays, a large number of trait observations have been synthesized by global networks (Gallagher et al., 2020). The TRY initiative (Kattge et al., 2011) was one of the most influential datasets with 2.3 billion trait data by the year 2021. Based on the TRY dataset, global LMA was estimated with upscaling techniques such as Bayesian modeling (Butler et al., 2017) (thereafter B2017) or the random forest model (Moreno-Martinez et al., 2018) (thereafter M2018). These advances in the retrieval of LMA provide the possibility to depict more accurate $O_3$ vegetation damage at the global scale.

Here, we present a major advance in large-scale assessments of $O_3$ plant damage using a trait-based approach. We implement LMA into a stomatal flux-based $O_3$ damage framework aiming at a unified representation of plant $O_3$ sensitivities over the global grids. We couple this new approach to the Yale Interactive terrestrial Biosphere (YIBs) model (Yue and Unger, 2015) and evaluate the derived $O_3$ sensitivities against observations. We further assess contemporary $O_3$ impacts on global GPP in combination with the recently developed LMA datasets (Butler et al., 2017; Moreno-Martinez et al., 2018;

Gallagher et al., 2020) (Fig. 1a) and the multi-model ensemble mean surface $O_3$ concentrations (Fig. 1b).
The updated risk map for $O_3$ vegetation damage is used to identify the regions and vegetation types most
at risk to $O_3$.

**2. Scheme development and calibration**
**2.1 The trait-based $O_3$ vegetation damage scheme**
We develop the new scheme based on the Sitch et al. (2007) (hereafter S2007) framework for transient
$O_3$ damage calculation. In the original S2007 scheme, the undamaged fraction $F$ for net photosynthetic
rate is dependent on the excessive area-based stomatal $O_3$ flux, which is calculated as the difference
between $f_{O3}$ and PFT-specific area-based threshold $y$, and modulated by the sensitivity parameter $a_{PFT}$:
$$F = 1 - a_{PFT} \times max\{f_{O3} - y, 0\} \tag{1}$$
where $a_{PFT}$ is calibrated and varies among PFTs with a typical range from "low" to "high" values
indicating uncertainties of plant species within the same PFT. The stomatal $O_3$ flux $f_{O3}$ (nmol m$^{-2}$ s$^{-1}$) is
calculated as:
$$f_{O3} = \frac{[O_3]}{r + \left[\frac{k_{O3}}{g_p \times F}\right]} \tag{2}$$
where $[O_3]$ is the $O_3$ concentration at the reference level (nmol m$^{-3}$), $r$ is the aerodynamic and boundary
layer resistance between leaf surface and reference level (s m$^{-1}$). $k_{O3}$ setting to 1.67 represents the ratio of
leaf resistance for $O_3$ to that for water vapor. $g_p$ represents potential stomata conductance for $H_2O$ (m s$^{-1}$

155 ).


Studies suggested that LMA could be used to unify the area-based plant sensitivities to $O_3$ (Li et al., 2016;
Feng et al., 2018), resulting in a constant mass-based parameter $a$ independent of plant species and PFTs:
$$a = a_{PFT} \times LMA \tag{3}$$
Here, we convert the area-based $O_3$ stomatal flux expression in Equation (1) to a mass-based flux as
follows:
$$F = 1 - a \times max\left\{\frac{f_{O3}}{LMA} - x, 0\right\} \tag{4}$$
where the new sensitivity parameter $a$ is a cross-species constant (nmol$^{-1}$ s g); $LMA$ is leaf mass per area
(g m$^{-2}$); the flux threshold is replaced by a mass-based value of $x$ (nmol g$^{-1}$ s$^{-1}$) (Feng et al., 2018).
Equations (2) and (4) can form a quadratic equation. The $F$ can be derived at each timestep (i.e. hourly)
and applied to net photosynthetic rate and stomatal conductance to calculate the O$_3$-induced damages.
The updated LMA-based framework (YIBs-LMA) reduces the number of O$_3$ sensitivity parameters from
three for each PFT (Sitch et al., 2007) in S2007 to a single parameter $a$ for all PFTs. For YIBs-LMA
framework, the default value of the $x$ threshold in Equation (4) is set to 0.019 nmol g$^{-1}$ s$^{-1}$ as recommended
by Feng et al. (2018).

**2.2 Dose-response relationship (DRR)**
We compare the simulated and observed sensitivities to O$_3$ so as to calibrate the LMA-based scheme. In
field experiments, DRR is used to quantify species-specific damage by O$_3$ with a generic format as
follows:
$R = 100 + S_O \times \phi_{O3}$                                                              (5)
where $R$ (%) is the relative percentage of a biotic indicator (such as biomass or yield) after and before O$_3$
damage; $\phi_{O3}$ is an area-based O$_3$ metric (e.g., POD$_y$ measured in sunlit leaves at the top of canopy); $S_O$
(usually negative) is the observed sensitivity derived as the slope of linear relationship between $R$ and
$\phi_{O3}$. We collected $S_O$ from DRRs with conventional criteria (typically POD$_{y=1}$ for natural PFTs and
POD$_{y=6}$ for crops as dose metrics (Clrtap, 2017); the biotic indicators include the relative biomass for
natural PFTs and relative yield for crops) among plant species from International Cooperative Programme
on Effects of Air Pollution on Natural Vegetation and Crops (CLRTAP) (Clrtap, 2017) and multiple
literature sources (Table S1). Such observations are used to calibrate the LMA-based scheme.

As a comparison with observations, we calculate annual relative GPP percentage ($R_{GPP}$, %) and $POD_y$ of
sunlit leaves in the first canopy layer (mmol m$^{-2}$ year$^{-1}$, based on per leaf area) from the vegetation model
to derive the slopes ($S_S$) of simulated DRRs. Here, $POD_y$ is a diagnostic variable calculated as:
$POD_y = \int \max\{f_{O3} - y, 0\}$                                                 (6)
where $f_{O3}$ represents the stomatal $O_3$ flux under instant $O_3$ stimulus at each timestep, which can be
calculated following Equation (2) at the leaf level; $y$ is the prescribed critical level (1 nmol m$^{-2}$ s$^{-1}$ for
natural or 6 nmol m$^{-2}$ s$^{-1}$ for crop species (Clrtap, 2017)). Excessive $O_3$ flux above y is accumulated for
the sunlit leaves of the top canopy layer and over the growing season to derive the $POD_y$. Simulated $S_S$ is
calculated as the slope of the regression between simulated $R_{GPP}$ (%) and $POD_y$ at the PFT level. Only the
dominant PFT in each grid is considered for the estimate of $S_S$ at both PFT-level or gridded analyses.

Similarly, mass-based $POD_x$ is derived from $O_3$-impacted $f_{O3}$ (nmol m$^{-2}$ s$^{-1}$) in Equation (2), together with
gridded LMA (g m$^{-2}$) and mass-based threshold $x$ (nmol g$^{-1}$ s$^{-1}$) as:
$$POD_x = \int \left( \frac{f_{O3}}{LMA} - x \right) \tag{7}$$

**2.3 Simulations and calibrations**
We perform two groups of supporting experiments (Table 1). The first group explores modeling
uncertainties associated with the mass-based framework: (1) YIBs-LMA_B2017 replaces the default
LMA map of M2018 (Moreno-Martinez et al., 2018) with B2017 (Butler et al., 2017). (2) YIBs-
LMA_PFT applies PFT-specific LMA values (Table S2) for each PFT without considering global LMA
geo-gradient. (3) YIBs-LMA_T replaces the default threshold of $x$=0.019 nmol g$^{-1}$ s$^{-1}$ with $x$ =0.006 nmol
g$^{-1}$ s$^{-1}$, which is an alternative parameter suggested by observations (Feng et al., 2018). The second group
of supporting experiments explores the differences between mass-based and S2007 area-based
frameworks. Typically, S2007 has a "low to high" $a_{PFT}$ range for each PFT. Here, a mean sensitivity
parameterization of S2007 (YIBs-S2007_adj) is re-calibrated according to $S_O$ in Table S1.

For all supporting experiments, the parameter $a$ for YIBs-LMA or the eight mean $a_{PFT}$ for YIBs-
S2007_adj are derived with the optimal 1:1 fitting between $S_S$ and $S_O$ to minimize the possible biases
(Tables 2 and S3-S6). The basic method for calibration is feeding the model with series values of $a$ or
$a_{PFT}$ until the predicted $O_3$ damage matches observations with the lowest normalized mean biases (NMB).
For all LMA-based experiments, $S_S$ from varied PFTs were grouped for the calibration of $a$, while for
$a_{PFT}$ in YIBs-S2007_adj, each $a_{PFT}$ is determined individually by matching simulated $S_S$ with $S_O$. Since
$S_O$ are available only for six out of the eight YIBs PFTs, including EBF, NF, DBF, $C_3$ grass, $C_4$ grass, and
crop (Table S1), $S_O$ of these PFTs are used for calibration. All runs are summarized in Table 1.

## 2.4 YIBs model and forcing data

In this study, all $O_3$ vegetation damage schemes are implemented in the YIBs model (Yue and Unger,
2015), which is a process-based dynamic global vegetation model incorporated with well-established
carbon, energy, and water interactive schemes. The model applies the same PFT classifications as the
Community Land Model (Bonan et al., 2003) (Fig. S1). Eight PFTs are employed including evergreen
broadleaf forest (EBF), needleleaf forest (NF), deciduous broadleaf forest (DBF), cold shrub (C_SHR),
arid shrubland (A_SHR), $C_3$ grassland (C3_GRA), $C_4$ grassland (C4_GRA), and cropland (CRO) (Fig.
S1). For each PFT, phenology is well-evaluated (Yue and Unger, 2015) to generate a reliable growing
season, which is crucial for the simulation of stomatal $O_3$ uptake (Anav et al., 2018). Photosynthesis and
stomatal processes are calculated using Farquhar et al. and Ball-Berry algorithms (Farquhar et al., 1980;
Ball et al., 1987), respectively. Leaf area index (LAI) and tree height are predicted dynamically based on
vegetation carbon allocation. The YIBs model has joined the multi-model ensemble project TRENDY
and showed reasonable performance in the simulations of global biomass, GPP, LAI, net ecosystem
exchange, and soil carbon relative to observations (Friedlingstein et al., 2020). Key plant biogeochemical
parameters of the YIBs model are adjusted for this research (Table S7).

The hourly modern-era retrospective analysis for research and applications version 2 (MERRA2) climate
reanalyses (Gelaro et al., 2017) are used to drive the YIBs model. The gridded LMA required for the main
mass-based simulation is derived from Moreno-Martinez et al. (2018) (M2018), which shows the highest
value of >150 g m$^{-2}$ for needleleaf forest at high latitudes while low values of ~40 g m$^{-2}$ for grassland and
cropland (Fig. 1a and Fig. S1). Grids with missing LMA data are filled with the mean of the corresponding
PFT. Contemporary $O_3$ concentration fields in the year 2010 from the multi-model mean in Task Force
on Hemispheric Transport of Air Pollutants (TF-HTAP) experiments (Turnock et al., 2018) (Fig. 1b) are
used as forcing data. The original monthly $O_3$ data are downscaled to hourly using the diurnal cycle
predicted by the chemistry-climate-carbon fully coupled model ModelE2-YIBs (Yue and Unger, 2015).
Generally, areas of severe $O_3$ pollution are found in the mid-latitudes of the Northern Hemisphere with
highest annual average $O_3$ concentration of over 40 ppbv in East Asia. All data are interpolated to the
spatial resolution of $1º \times 1º$.

**3. Results**
**3.1 Comparison of simulated sensitivities with observations**
Simulated relative GPP percentage ($R_{GPP}$) at global grids were sorted by dominant PFTs (Fig. S1) and
plotted against area-based accumulated phytotoxic $O_3$ dose above a threshold y nmol $m^{-2}$ $s^{-1}$ ($POD_{y=1}$) at
the corresponding grids (Fig. 2). The DRR shows varied slopes among different PFTs, resulting in a
coefficient of determination ($R^2$) around 0.54 for all PFTs (Figs 2a-2c). We further calculated the mass-
based accumulated phytotoxic $O_3$ dose above a threshold of 0.019 nmol g $s^{-1}$ ($POD_{x=0.019}$) and compared
it with $R_{GPP}$. The updated DRR showed convergent slopes and reached a high $R^2$ of 0.77 across all PFTs
(Figs 2d-2f), suggesting that the mass-based scheme could better unify $O_3$ sensitivities among different
PFTs.

We then calibrated the single, best-fit *a* value for the YIBs-LMA framework by minimizing the absolute
difference between simulated ($S_S$) and observed ($S_O$) slopes of $O_3$ DRR for all PFTs. With different *a*
parameters, the YIBs-LMA framework yielded considerably high $R^2$ of ~1.0 but varied biases between
simulated and observed $O_3$ impacts across PFTs (Fig. 3). Both the 1:1 fitting and the lowest bias between
$S_S$ and $S_O$ were achieved with an optimal *a* =3.5 $nmol^{-1}$ s g (Fig. 3c). Notably, such calibration of *a* is
robust under different $O_3$ fields (see Fig. S2). Consistent with observations, YIBs-LMA with this optimal
*a* parameter simulated low $S_S$ of -0.18% and -0.36% per mmol $m^{-2}$ $year^{-1}$ of $POD_{y=1}$ for evergreen
broadleaf forest and needleleaf forest, respectively (Figs 4a, b), median $S_S$ from -0.53% per mmol $m^{-2}$
$year^{-1}$ for arid shrubland (Fig. 4e), and high $S_S$ from -0.64% to -1.04% per mmol $m^{-2}$ $year^{-1}$ for deciduous
broadleaf forest, $C_3$/$C_4$ grassland, cropland and cold shrubland (-3.28% for crops with $POD_{y=6}$, Figs 4c-
d, 4f-h).

**3.2 Global map of $O_3$ vegetation damage**
We estimated contemporary GPP reductions induced by $O_3$ with the global concentrations of surface $O_3$
(Fig. 1b) in the year 2010. The YIBs-LMA framework using an increase of $a$ parameter yielded an almost
linear enhancement of global GPP reduction (Fig. S3) with consistent spatial distributions (Fig. S4). The
simulation with the optimal $a$ =3.5 nmol$^{-1}$ s g predicted a global GPP reduction of 4.8% (Fig. 5a), which
was similar to the value estimated with the area-based S2007 scheme (YIBs-S2007_adj, Table 1). Large
reductions of >10% were predicted over the eastern U.S., western Europe, eastern China, and India (Fig.
5a). Hotspots were mainly located in cropland and agricultural areas mixed with deciduous broadleaf
forest or grassland, accompanied by moderate to high levels of surface $O_3$. Few discrepancies between
the damage maps of YIBs-LMA and YIBs-S007_adj were found (Fig. 5b and Fig. S5), even though the
number of parameters was greatly reduced in the YIBs-LMA scheme.

For YIBs-LMA, PFTs with low LMA such as cropland, grassland, and deciduous broadleaf forest account
for 73.3 Pg C yr$^{-1}$ (50.0%) of the global GPP (Table S8). However, these PFTs contributed to a total GPP
reduction of 5.4 Pg C yr$^{-1}$ (75.5% of total GPP loss) by $O_3$ damage. In contrast, evergreen broadleaf and
needleleaf forests with high LMA accounted for 48.8 Pg C yr$^{-1}$ (33.0%) of total GPP but yielded only a
reduction of 0.75 Pg C yr$^{-1}$ (10.5% of total GPP loss). Differences in GPP percentage losses were in part
associated with the global pattern of $O_3$ concentrations, which were usually higher over mid-latitudes with
populated cities and dense crop plantations (Fig. 1b). However, the differences in LMA and simulated $O_3$
sensitivities of these PFTs also made important contributions to such discrepancies in GPP damage.

**3.3 Uncertainties of the LMA-based scheme**
We quantified the uncertainties in the LMA-based scheme by comparing simulated GPP damages among
different experiments (Table 1). The experiment with the alternative LMA map of B2017 (Fig. S6)
showed similar spatial patterns but a slightly enhanced GPP reduction of 5.3% (Fig. 6a) compared to the
simulations using LMA map of M2018 (Fig. 5a). The B2017 map contains much LMA data than M2018
(~40%), leading to unexpected high $O_3$ threats over the tundra in the Arctic (Fig. S7). Another experiment
using PFT-specific LMA estimated a global GPP reduction of 4.6% (Fig. 6b) with a consistent spatial
pattern as the prediction in YIBs-LMA, suggesting that the PFT-level LMA can be used in case of the
lack of regional LMA data. The third experiment with an alternative threshold flux (Feng et al., 2018) of
0.006 nmol g$^{-1}$ s$^{-1}$ estimated a high GPP reduction of 6.5% (Fig. 6c) due to the overestimations of O$_3$
sensitivities for some tree PFTs (Fig. 7). The fourth run, YIBs-S2007_adj, predicted a similar global GPP
damage of 4.8% as the YIBs-LMA run with a high spatial correlation coefficient of 0.98 (Fig. 6d). Such
good consistency is mainly due to the application of recalibrated PFT-level sensitivities in YIBs-
S2007_adj. Finally, we tested a new calibration excluding CRO, the PFT that contributed the most to the
calibration biases (shown as orange dashed lines in Fig. S8). The results gave an optimal *a* of 3.2, with
global damage of 4.5%. All sensitivity experiments achieved consistent results as the YIBs-LMA
simulation with damages ranging from 4.5% to 6.5% and spatial correlation coefficients larger than 0.94.

**4. Discussion**
**4.1 Mechanisms behind the LMA-based approach**
In recent decades, the plant science community examined how traits could be used to differentiate and
predict the function of plant species (Reich et al., 1997; Reich et al., 1999). LMA, related to leaf density
and thickness, is a key trait reflecting many aspects of leaf function (Reich et al., 1998). In the field of O$_3$
phytotoxicology, experiments have revealed plants with high LMA usually have thick leaves with
physical and chemical defenses (Poorter et al., 2009), which can strengthen their resistance to O$_3$ (Li et
al., 2016; Feng et al., 2018). On the contrary, plants with low LMA normally have thin leaves which are
likely to be less O$_3$-tolerant (Li et al., 2016; Feng et al., 2018). Moreover, it seems plausible that the
oxidative stress caused by a given amount of stomatal O$_3$ flux per unit leaf area would be distributed over
a larger leaf mass, and hence diluted, in a leaf with high LMA. Such an LMA-O$_3$ sensitivity relationship
can be well reproduced by our LMA-based model (Figs 8a and 8b). Below we explore the linkage between
O$_3$ plant sensitivities and the mutual adaptation of growth strategies and leaf morphology with plant leaf
trade-off theory (Reich et al., 1999; Shipley et al., 2006).

In the natural world, plants often adapt to maximize carbon uptake under prevailing conditions (Reich et
al., 1998; Shipley et al., 2006). To make full use of resources in the growing season, leaves under varied
living conditions choose either fast photosynthetic rates (fast-growing deciduous types) or long

photosynthesis duration (slow-growing evergreen types) with compatible leaf structures (Reich, 2014; Diaz et al., 2016). The former species expand leaf area (low LMA) to maximize light interception while the latter species produce thick and mechanically strong leaves (high LMA) with ample resistant substances for durable utilization (Poorter et al., 2009) in resource-limited and/or environment-stressed habitats (Wright et al., 2002). As a side effect of such leaf trade-offs, deciduous plants with their high rates of photosynthesis, associated large stomatal conductance (Davison and Barnes, 1998; Henry et al., 2019), and less total defense capacity through the leaf profile (Poorter et al., 2009), are highly $O_3$ sensitive (Mode1 in Fig. 9). In contrast, the moderate photosynthesis, relatively low maximum stomatal conductance (Davison and Barnes, 1998; Henry et al., 2019), and reinforced dense leaves (Poorter et al., 2009) lead to low sensitivity for evergreen plants (Mode2 in Fig. 9). Therefore, in our modeling practice, the mass-based $O_3$ gas exchange algorithm can be regarded as taking into account several interrelated factors such as growth-driven gas exchange requirements, gas path length, and biochemical reserves, in a unified, simplified and effective manner via LMA.

## 4.2 Implication of potential risks for fast-growing plants

Our new approach reflected the general experimental findings that deciduous plants are much more vulnerable to $O_3$ than evergreen species (Li et al., 2017; Feng et al., 2018), and in turn within a PFT, early-successional/pioneers with low LMA are likely more vulnerable than late-successional/canopy trees with high LMA (Fyllas et al., 2012). This law has been neglected in previous modeling studies due to the dependence on the limited observed data used for PFT-specific tuning. Our LMA-based approach bridges this gap through grid-based parameterization, and in addition, our data-model integration specifically emphasizes the broad high risks for fast-growing plants, especially for crops. Among PFTs, crops may endure the largest $O_3$ threats (Davison and Barnes, 1998; Feng et al., 2021; Mukherjee et al., 2021) because they are artificially bred with high photosynthetic capacities (Richards, 2000), stomatal conductance, generally low LMA (Bertin and Gary, 1998; Wang and Shangguan, 2010; Wu et al., 2018; Li et al., 2018) (roughly 30-60 g m$^{-2}$), and cultivated in populated regions with high ambient $O_3$ concentrations. Modern technology aims to promote crop yield (Herdt, 2005), but this can potentially elevate crop sensitivities to $O_3$ (Biswas et al., 2008; Biswas et al., 2013). This study estimated the highest

annual mean GPP damage for crop, 12.6%, which is at the high end of the 4.4-12.4% of the $O_3$-induced
yield loss estimated for global modeling of soybean, wheat, rice, and maize (Mills et al., 2018a).
Furthermore, human-induced land use activities may also increase $O_3$ damage risks. The global demand
for food and commodities leads to the conversion of natural forests to irrigated croplands, grazing
pastures, and economical-tree plantations (Curtis et al., 2018; Zalles et al., 2021). Meanwhile, the urgent
actions to combat climate change promote large-scale afforestation and reforestation (Cook-Patton et al.,
2020). These land use changes with fast-growing plant species may increase the risks of terrestrial
ecosystems to surface $O_3$.

**4.3 Advances in the global $O_3$ damage assessment**
For the first time, we implemented plant trait LMA into a process-based $O_3$ impact modeling scheme and
obtained reasonable interspecific and inter-PFT $O_3$ responses supported by observations. The similarity
between YIBs-S2007 and YIBs-LMA shown in Fig. 5 revealed an advance in the modeling strategy.
Simulated $O_3$ damage in YIBs-S2007 is based on the PFT-level calibrations that tuned sensitivity
parameters of each PFT with observed DRRs. Such refinement is a data-driven approach without clear
physical reasons. Instead, the YIBs-LMA framework converts the area-based responses to mass-based
ones and achieves better unification in $O_3$ sensitivities among different PFTs. In this algorithm, the $O_3$
damage efficiency is inversely related to plant LMA, which influences both the $O_3$ uptake potential and
the detoxification capability of the vegetation. The similarity in the global assessment of $O_3$ vegetation
damage between YIBs-S2007 and YIBs-LMA further demonstrated the physical validity of LMA-based
scheme in the Earth system modeling, because the independent LMA map was applied in the latter
approach.

In addition to the advance in physical mechanisms, the LMA-based approach improves global $O_3$ damage
assessments in the following aspects. First, it significantly reduces the number of required key parameters.
To account for interspecific sensitivities, many schemes have to define PFT-level parameters to capture
the ranges of plant responses (Sitch et al., 2007; Felzer et al., 2009; Lombardozzi et al., 2015). As a result,
those schemes rely on dozens of parameters which increases the uncertainties of modeling and the

difficulties for model calibration. The LMA-based approach requires the calibration of one single parameter *a*, largely facilitating its application across different vegetation models. Second, the new approach accounts for the continuous spectrum of $O_3$ sensitivities. Previous studies usually categorized species into groups of low or high $O_3$ sensitivity, depending on very limited data from $O_3$ exposure experiments. As a result, gridcells for a specific PFT share the same sensitivities regardless of their geographic locations and ecosystem characteristics. In reality, there are hundreds and thousands of plant species in each PFT and they usually have large variations in biophysical parameters including LMA and $O_3$ sensitivities. The LMA-based approach takes advantage of the newly revealed unifying concept in $O_3$ sensitivity (Li et al., 2016; Feng et al., 2018; Li et al., 2022) and the recent development in a trait-based LMA global map (Fig. 1a). Such configurations present a spectrum of gridded $O_3$ sensitivities (Fig. 8a) following the variations of LMA distribution.

## 4.4 Outlook for future modeling

In nature, all aspects of plant physiochemical processes, such as growth, development, reproduction, and defense, are influenced by abiotic factors like water availability, temperature, $CO_2$ concentration, and light resources (Kochhar and Gujral, 2020). In our modeling, the cumulative $O_3$ fluxes are based on dynamic plant simulations with well-established DGVM to calculate the effects of these abiotic factors. LMA is considered as a factor representing the vulnerability of each species, by which divergent responses to the same $O_3$ stomatal dose can be further differentiated. In fact, many other key variables in DGVMs, for example, leaf photosynthetic traits ($V_{cmax}$ and $J_{max}$), nutrient traits (leaf nitrogen and phosphorus), morphological traits (leaf thickness and size), and phenology-related traits (leaf life span) are all more or less interlinked with LMA (Walker et al., 2014). There are some generic regression relationships between them. In addition, efforts are being made to directly predict key traits, including LMA, through environmental factors. As a result, considerable improvements can be made in the direction of trait-flexible modeling within the existing DGVM frameworks. Our study demonstrates the validity of LMA-based approach for the $O_3$ plant damage modeling.

Although we used the most advanced LMA integrated from available observations, this dataset was developed based on static global grids and revealed the mean state for each pixel. In reality, LMA can vary with biotic/abiotic factors like leaf position in the canopy (Keenan and Niinemets, 2017), phenology, plant health, living environment (Fritz et al., 2018), and climate (Wright et al., 2005; Cui et al., 2020). Even long-term exposure to $O_3$ can alter leaf morphological characteristics and LMA (Li et al., 2017). In future studies, simulations from local to global scales could implement the spatiotemporal variations in LMA taking into account the demographic information and environmental forcings. We expect a breakthrough in the calculation of reliable LMA to achieve fully dynamic predictions of $O_3$ plant damage in Earth System Modeling, thus facilitating the research of plant response and adaption in changing environments.

**Code availability**

The codes of YIBs model with LMA-based $O_3$ damaging scheme are shared at https://zenodo.org/record/6348731.

**Data availability**

Results of all simulations (listed in Table 1) are available upon request. Data for Figures in the main article are shared at https://zenodo.org/record/6348731. The global maps of specific leaf area (SLA) to derive LMA for M2018 and B2017 are from https://www.try-db.org/TryWeb/Data.php#59 and https://github.com/abhirupdatta/global_maps_of_plant_traits, respectively. Monthly $O_3$ data is from https://doi.org/10.5194/acp-18-8953-2018. Calibration data are summarized in Table S1.

**Author Contributions**

X.Y., S.S. and N.U. designed the research, Y.M.M. performed modeling, data analyses, virtualization and wrote the draft. J.U, L.M., Z.Z.F, and A.W.C advised on concepts and methods. C.G. helped write draft. H.Y.Y., M.C.D.R helped with coding. H.Z., C.G.T., Y.C., Y.D.L., and Y.S.X. helped with data collection. All authors commented and revised the manuscript.

## Competing interests

The authors declare no conflict of interests.

## Financial support

Xu Yue acknowledges funding supports from the National Natural Science Foundation of China (grant no. 42293323) and Jiangsu Science Fund for Distinguished Young Scholars (grant no. BK20200040). Yimian Ma acknowledges financial support from China Scholarship Council (CSC no. 201804910712). Johan Uddling acknowledges the strategic research area Biodiversity and Ecosystems in a Changing Climate, BECC. SS, NU, LM, AC were supported by NERC funding (NE/R001812/1).

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

Table 1. Summary of simulations.

| Experiment [a] | Method | Thresholds [a] ($x$ or $y$) | LMA format | LMA map | Optimal ($a$ or $a_{PFT}$) | Tests ($a$ or $a_{PFT}$) |
|---|---|---|---|---|---|---|
| YIBs-LMA | | $x$=0.019 | gridded | M2018 | $a$=3.5 (Table 2) | five tests ($a$=2.5, 3, 3.5, 4, 4.5) |
| YIBs-LMA_PFT | Mass-based | $x$=0.019 | PFT-specific | M2018 | $a$=2.0 (Table S3) | five tests ($a$=2, 2.5, 3, 3.5, 4) |
| YIBs-LMA_T | | $x$=0.006 | gridded | M2018 | $a$=3.0 (Table S4) | five tests ($a$=2, 2.5, 3, 3.5, 4) |
| YIBs-LMA_B2017 | | $x$=0.019 | gridded | B2017 | $a$=2.8 (Table S5) | five tests ($a$=2, 2.5, 2.8, 3, 3.5) |
| YIBs-S2007_adj | Area-based | 8 values for $y$ (Table S6) | / | / | 8 values for $a_{PFT}$ (Table S6) | 40 tests (five each for 8 PFTs) |

[a] Units of thresholds are nmol g$^{-1}$ s$^{-1}$ for $x$ and nmol m$^{-2}$ s$^{-1}$ for $y$

[b] Units of key parameters are nmol$^{-1}$ s g for $a$ and nmol$^{-1}$ m$^2$ s for $a_{PFT}$

**Table 2.** Calibrations of the YIBs-LMA [a] experiment with varied $a$.

| PFT | $S_O$ | $S_S$ | | | | | $S_S/S_O$ [b] | | | | |
|---|---|---|---|---|---|---|---|---|---|---|---|
| | | a=2.5 | a=3.0 | **a=3.5** | a=4.0 | a=4.5 | a=2.5 | a=3.0 | **a=3.5** | a=4.0 | a=4.5 |
| EBF | -0.19 | -0.13 | -0.16 | **-0.18** | -0.21 | -0.23 | 0.70 | 0.83 | **0.96** | 1.08 | 1.20 |
| NF | -0.23 | -0.26 | -0.31 | **-0.36** | -0.40 | -0.45 | 1.14 * | 1.35 * | **1.56 *** | 1.75 * | 1.95 * |
| DBF | -0.70 | -0.51 | -0.60 | **-0.69** | -0.78 | -0.87 | 0.72 | 0.86 | **0.99** | 1.12 | 1.24 |
| C_SHR | / | -0.75 | -0.90 | **-1.04** | -1.18 | -1.31 | / | / | / | / | / |
| A_SHR | / | -0.38 | -0.45 | **-0.53** | -0.60 | -0.66 | / | / | / | / | / |
| C4_GRA | -0.85 | -0.71 | -0.84 | **-0.97** | -1.10 | -1.22 | 0.83 | 0.99 | **1.14** | 1.29 | 1.44 |
| C3_GRA | -0.62 | -0.47 | -0.55 | **-0.64** | -0.73 | -0.81 | 0.75 | 0.89 | **1.03** | 1.17 | 1.30 |
| CRO | -3.35 | -1.97 | -2.57 | **-3.28** | -4.11 | -5.10 | 0.59 | 0.77 | **0.98** | 1.23 | 1.52 |
| Fitting [c] | / | 0.61 | 0.79 | **0.99** | 1.23 | 1.50 | / | / | / | / | / |
| Median | / | / | / | / | / | / | 0.74 (0.72) | 0.88 (0.86) | **1.01 (0.99)** | 1.20 (1.17) | 1.37 (1.30) |
| Std | / | / | / | / | / | / | 0.19 (0.09) | 0.21 (0.08) | **0.23 (0.07)** | 0.25 (0.08) | 0.28 (0.13) |


[a] All runs from the YIBs-LMA experiment use $x$=0.019 nmol g$^{-1}$ s$^{-1}$ and LMA map from M2018.
[b] Slopes of simulated DRRs ($S_S$) are divided by observations ($S_O$, Table S1) to derive the model-to-observation ratios ("$S_S/S_O$").
$O_3$ dose metric is $POD_{y=1}$ for natural PFTs and $POD_{y=6}$ for crops. The Median and standard deviation (Std) of $S_S/S_O$ ratios of
all PFTs are calculated for each set of specific parameter $a$. The values in parentheses exclude the effect of those numbers
marked with * that are out of 1 standard deviation.
[c] The slopes (Fitting) of linear regressions between $S_O$ and $S_S$ are listed for each $a$. The optimal $a$ with 1:1 fitting between $S_S$
and $S_O$ is bolded.

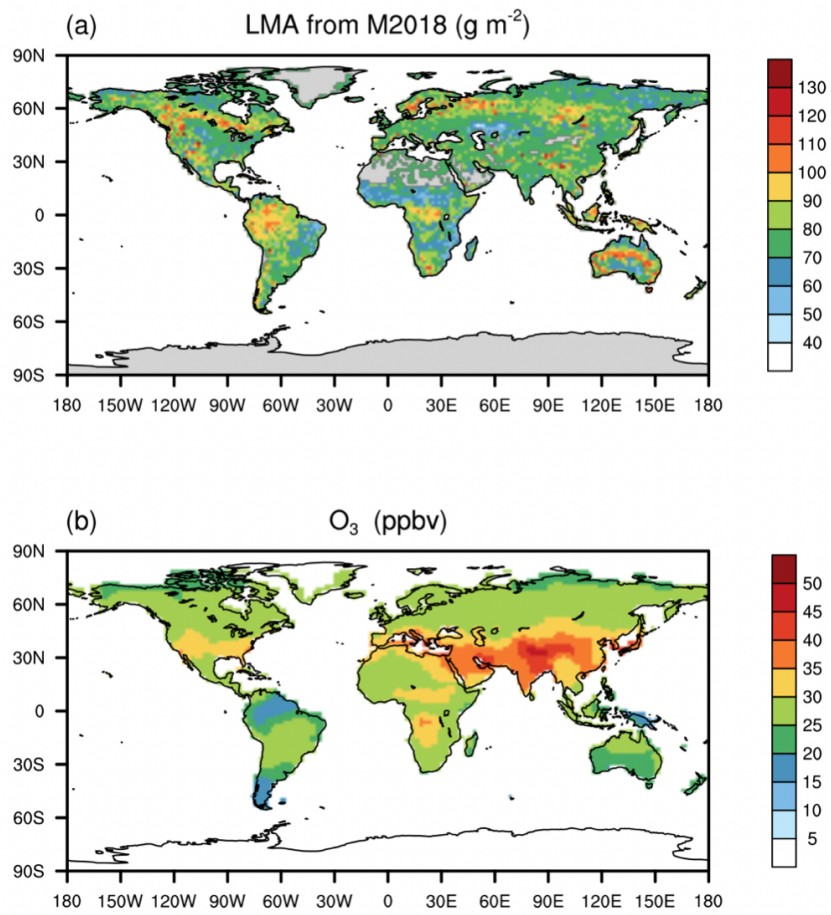


**Figure 1.** Global leaf mass per area (LMA) and surface ozone (O$_3$) concentrations. The (a) LMA is
adopted from Moreno-Martinez et al. (2018) (M2018) and (b) annual mean O$_3$ is derived from TF-HTAP
(Turnock et al., 2018).



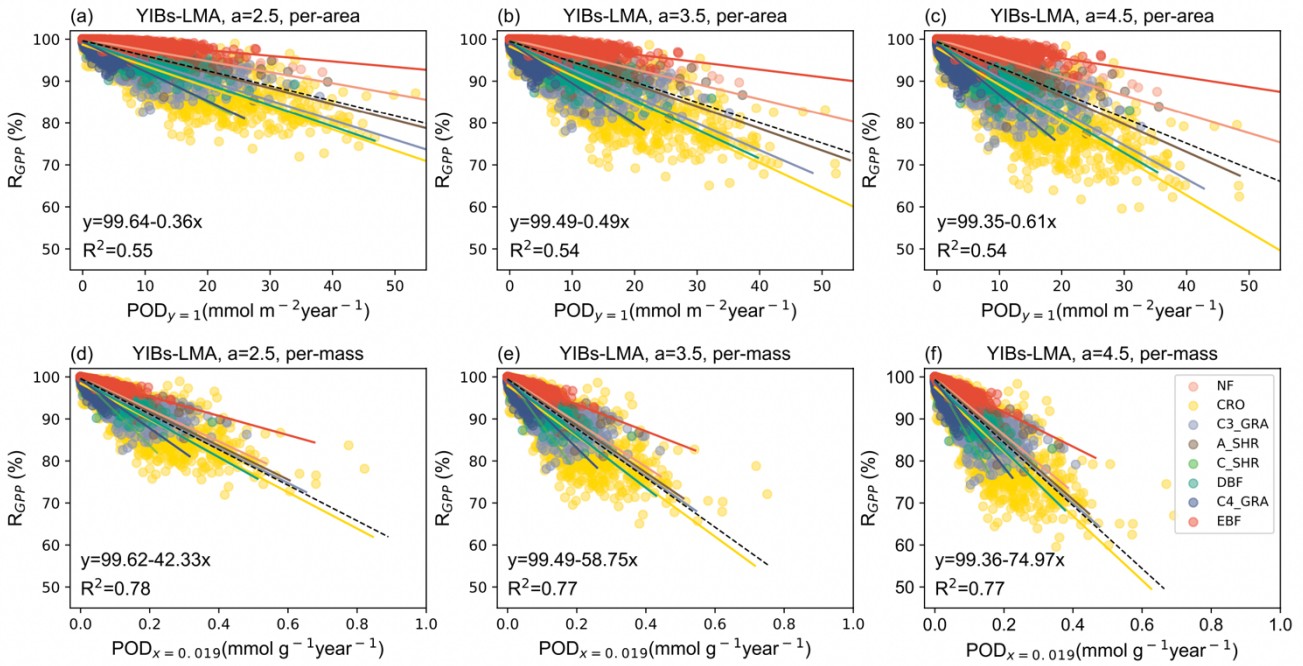


**Figure 2.** Simulated area-based (top) or mass-based (bottom) DRRs for the YIBs-LMA experiment. Three tests from the YIBs-LMA experiment all adopt $x$=0.019 nmol g$^{-1}$ s$^{-1}$ and gridded LMA from M2018 with parameter $a$=2.5, 3.5, 4.5 nmol$^{-1}$ s g, respectively. Each dot represents estimated POD-R$_{GPP}$ (POD$_{y=1}$ for (a)-(c), POD$_{x=0.019}$ for (d)-(e)) at a grid with corresponding PFT. The PFT-specific regressions between area- or mass- based POD and R$_{GPP}$ are displayed with solid lines shown in legend. Regression relationships of all PFTs are displayed in black dash line with coefficients of determination (R$^2$) denoted on each panel. Note the differences of ranges in x axis among PFTs. The YIBs-LMA experiment is summarized in Table 1.

697

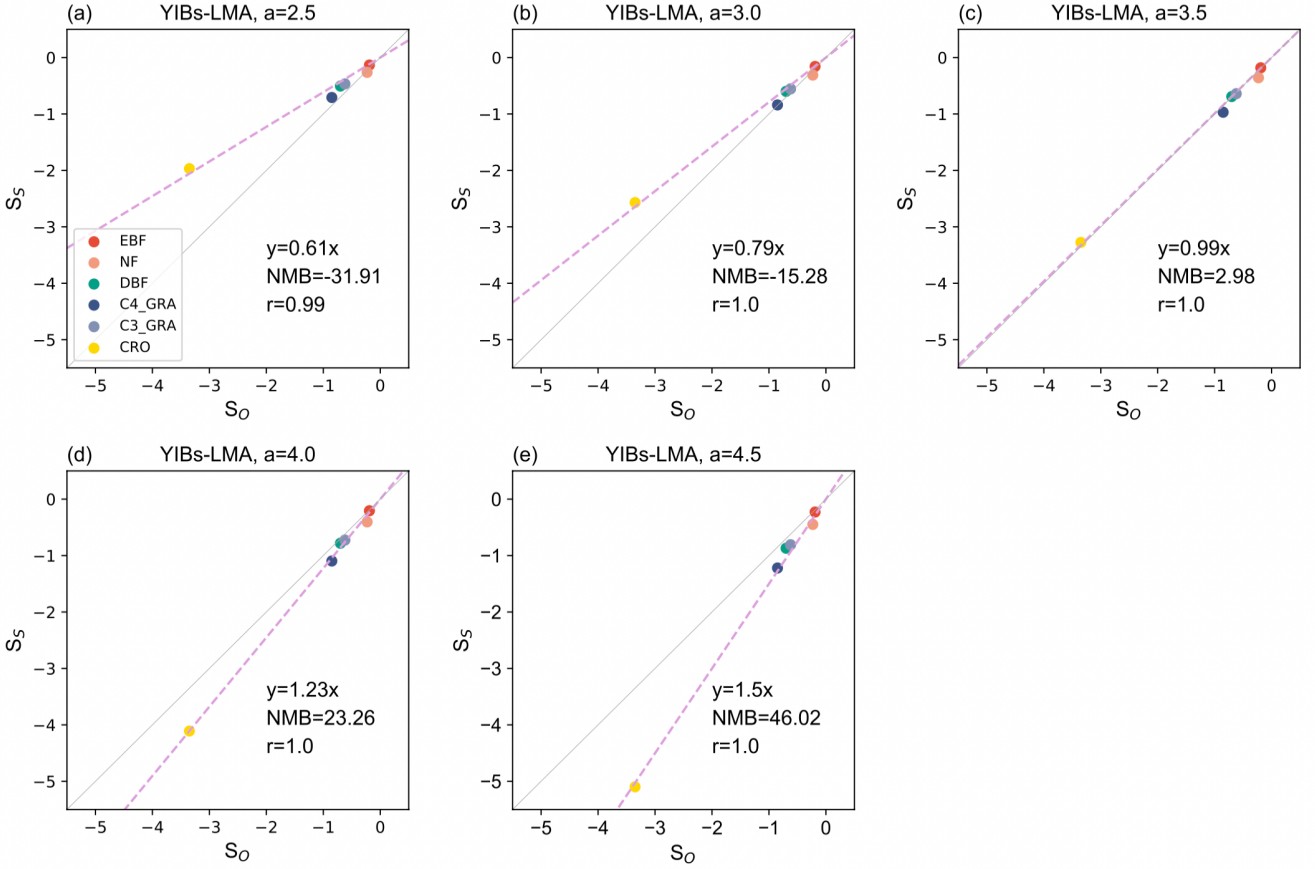

**Figure 3.** Comparison between $S_O$ (% per mmol m$^{-2}$) and $S_S$ (% per mmol m$^{-2}$) for the YIBs-LMA experiment. Five tests from the YIBs-LMA experiment all adopt $x$=0.019 nmol g$^{-1}$ s$^{-1}$ and gridded LMA from M2018 but with varied parameter $a$ from (a) 2.5 to (e) 4.5 nmol$^{-1}$ s g. $S_O$ are from Table S1. $S_S$ are derived as the slope between $R_{GPP}$ and $POD_y$. The linear regression (dashed lines), 1:1 fitting (light grey lines), normalized mean biases (NMB), and correlation coefficient (r) between $S_S$ and $S_O$ for varied PFTs are shown on each panel. The $S_S$ and $S_O$ of CRO are derived using $POD_{y=6}$ while other PFTs use $POD_{y=1}$. The YIBs-LMA experiment is described in Table 1.

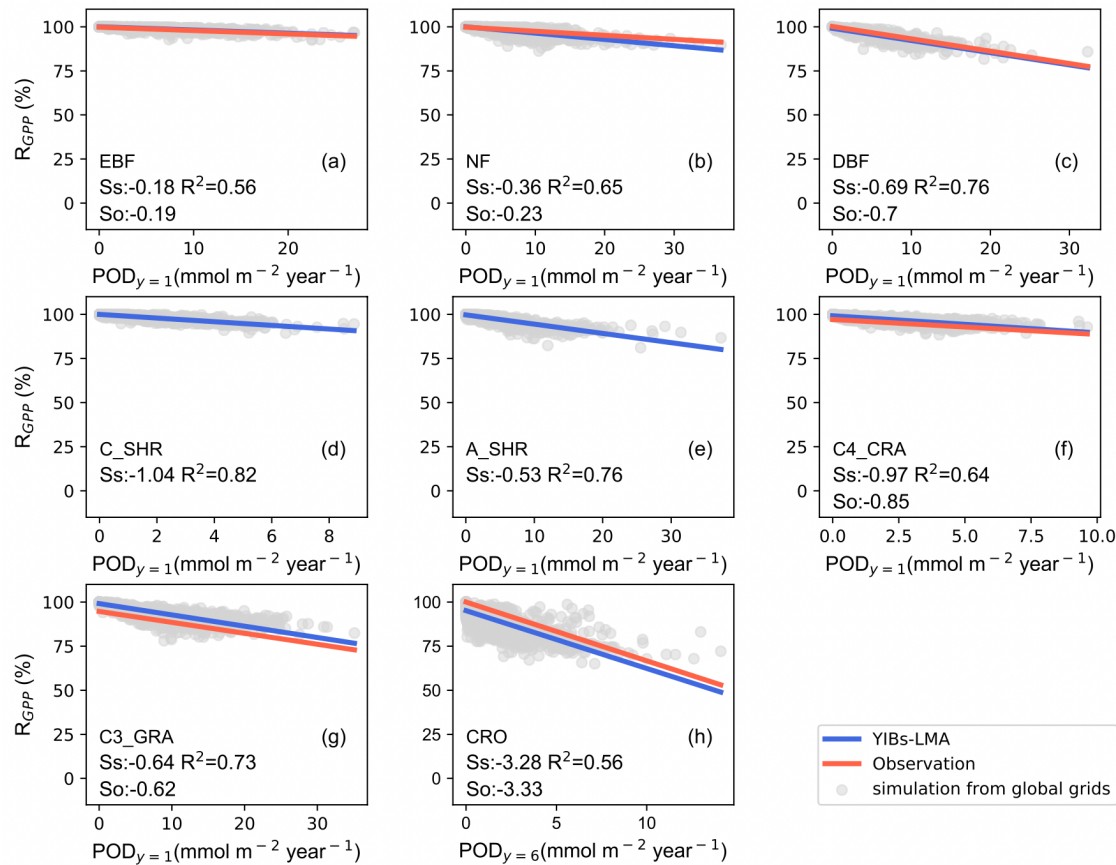

708

**Figure 4.** Comparisons of observed and simulated dose-response relationships. Simulated PFT-specific DRRs are derived from YIBs-LMA with gridded LMA from M2018, $x$=0.019 nmol g$^{-1}$ s$^{-1}$, and $a$=3.5 nmol$^{-1}$ s g. Each dot represents results from a gridcell with corresponding PFT. The regressions between relative GPP percentage ($R_{GPP}$) and leaf area-based stomatal O$_3$ uptake fluxes (POD$_{y=1}$ for natural PFTs and POD$_{y=6}$ for crops) are shown for observations (red, see Table S1) and simulations (blue) with slopes of DRRs denoted as $S_o$ and $S_s$, respectively. $S_O$ are missing for (d) cold and (e) arid shrubs. Coefficients of determination ($R^2$) of simulations are displayed in each panel. Note the differences of ranges in x axis among PFTs (PFTs are shown in Fig. S1).

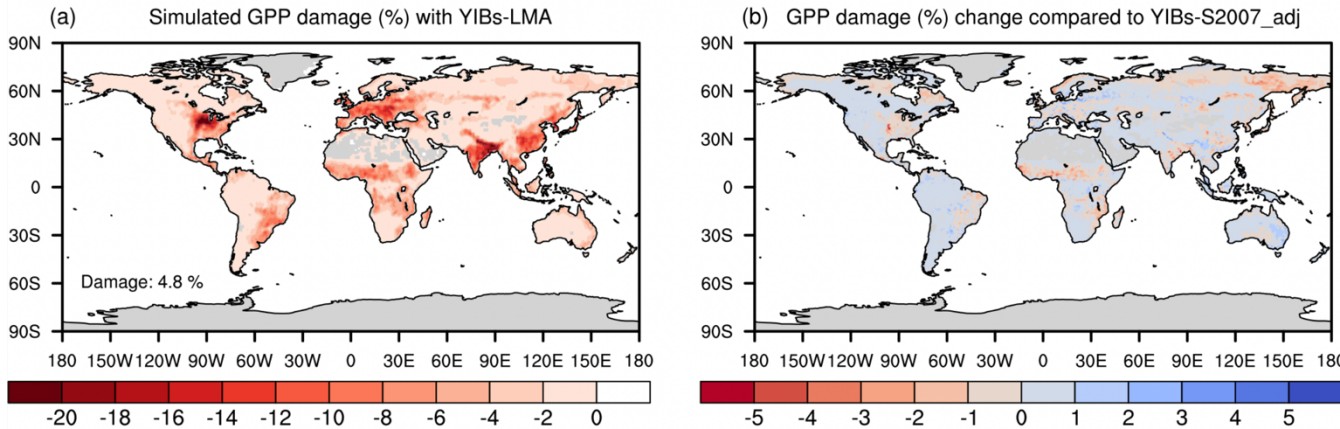

**Figure 5.** Global O$_3$ vegetation damage simulated with the LMA-based scheme. Results shown are the (a) GPP reduction percentages by O$_3$ simulated with the YIBs-LMA framework (gridded LMA from M2018, $x$=0.019 nmol g$^{-1}$ s$^{-1}$, and $a$=3.5 nmol$^{-1}$ s g), and (b) their differences compared to the predictions from YIBs-S2007_adj simulation. Blue (red) patches indicate the regions where damages predicted in YIBs-LMA are lower (higher) than those in YIBs-S2007_adj.

724

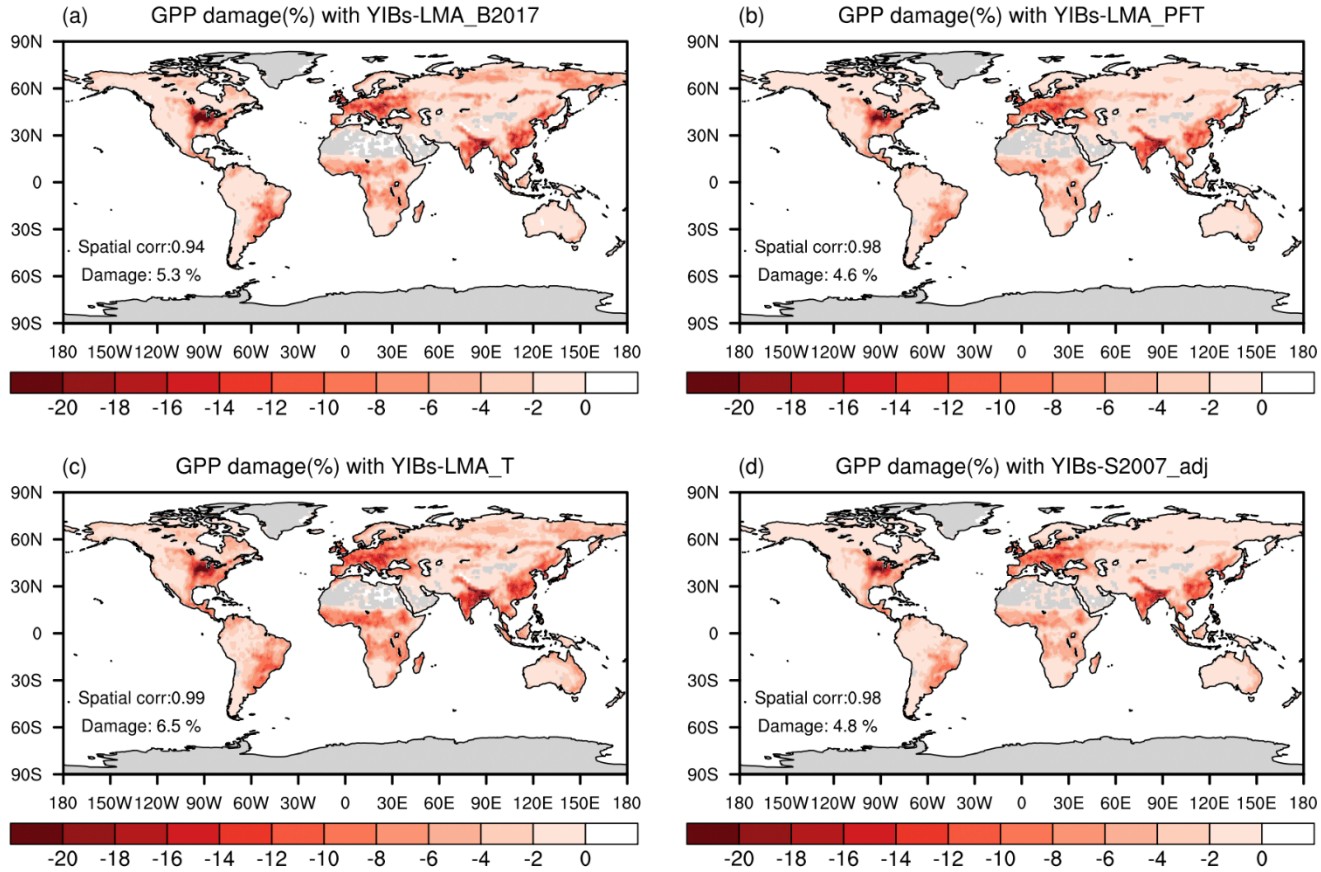

725

**Figure 6.** Global O$_3$-induced GPP reductions simulated in four supporting experiments. All damage maps are based on optimal parameters shown in Table 1. The spatial correlation coefficients between YIBs-LMA and these supporting simulations are shown on each panel as well as the global average damage percentage of each map. Simulations are described in Table 1.


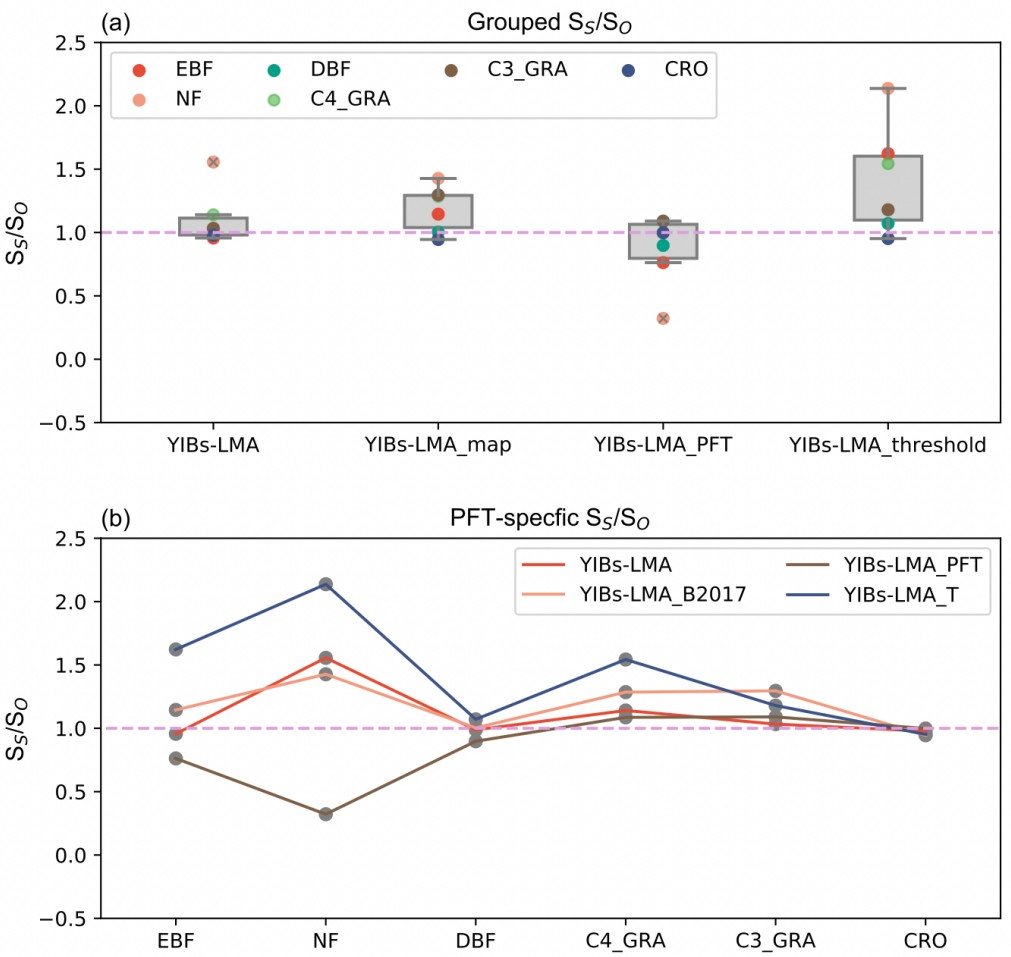


**Figure 7.** Comparison of $S_S/S_O$ among supporting experiments. Individual ratios for (b) different PFTs

are grouped to the boxplot in (a). All experiments adopt optimal parameters shown in Table 1.


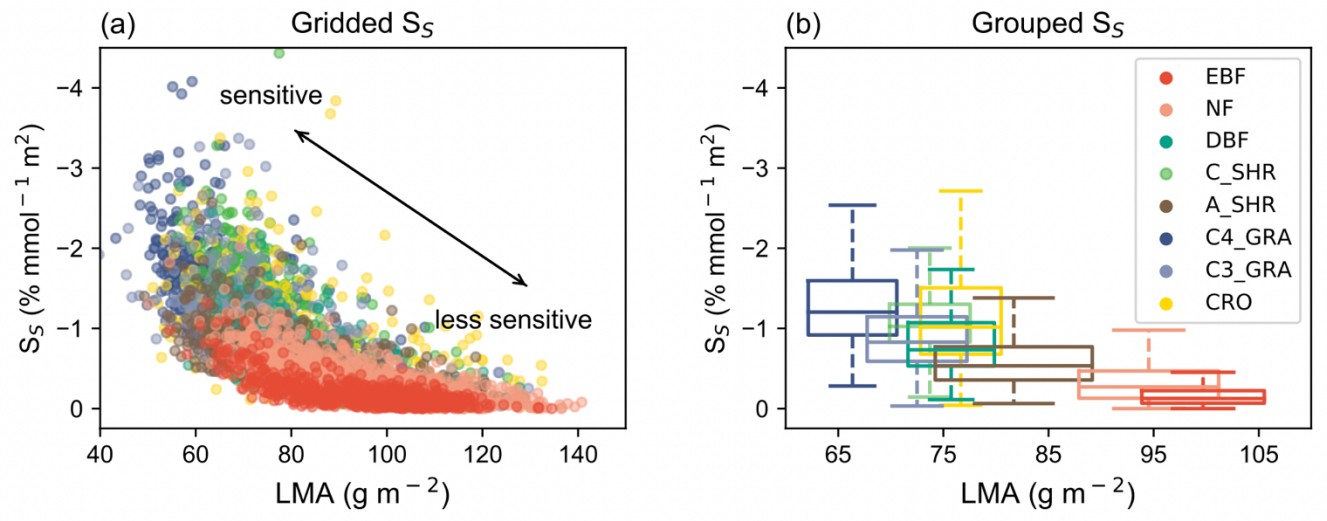


**Figure 8.** Relationships between $O_3$ sensitivity and LMA. (a) Simulated $O_3$ sensitivity ($S_S$) at each grid is
compared with LMA for different PFTs. Gridded $S_S$ is derived as GPP change per unit $POD_{y=1}$ from the
YIBs-LMA simulation. Each point represents the value in a grid cell with a dominant PFT. (b) The PFT-
level relationships between LMA and $O_3$ sensitivity are grouped as boxplots, which indicate the median,
$25^{th}$ percentile, and $75^{th}$ percentile of y-axis variables within the same PFT. The width of boxplots
represents one standard deviation of LMA for a specific PFT.

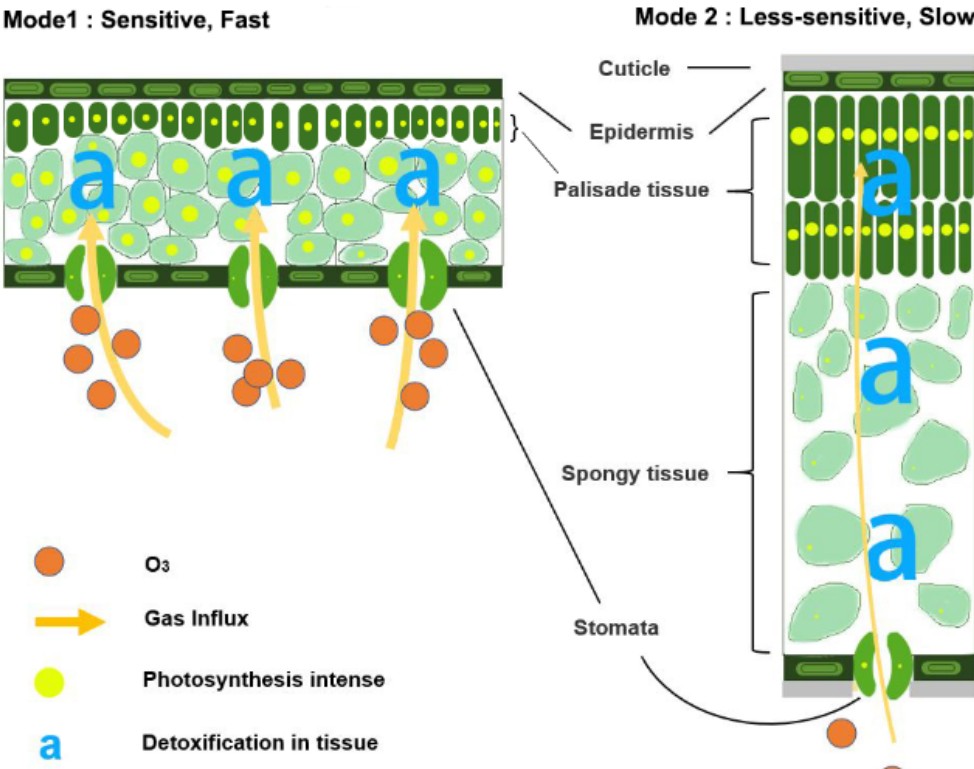


**Figure 9.** Illustration of the relationships between leaf trade-off strategy and its sensitivity to $O_3$