# Peer review of "Implementation of trait-based ozone plant sensitivity in the Yale"

_Geoscientific Model Development, 2022_

## Author Comment (AC1)

**Response to the review comment 1 on gmd-2022-227**

*Review for GMD-2022-227 by Yimian Ma et al.*

*------------------------------------------*

*The authors present a submodel for ozone damage to different plant types inserted into an existing biosphere model (YIBs). This new ozone model calculates plant damage, expressed as GPP penalties, based on a unified sensitivity interacting with leaf mass per area. They conclude that approx. 5% of global GPP is not materialized due to ozone damage.*

*I perceive the study as a major advance over previous approaches to model ozone damage to plants, taking into account latest findings on leaf mass rather than area as defining factors for ozone sensitivity across plant types. The manuscript is well written and the steps taken to develop and integrate the ozone model into YIBs are sound. All conclusions are grounded on the presented evidence. Nonetheless, I see several places where the study could be amended; they are detailed in the following. While I strongly suggest to consider these, none of them questions the relevance and overall validity of the approach, though.*

*In conclusion, I recommend a major revision of the article. The 'major' is a sum of many 'minor' elements. If the authors are able to address my concerns, I clearly support a publication of this article.*

Response: thank you very much for your helpful comments and constructive suggestions for further improving our manuscript. We have carefully considered all comments and revised our manuscript accordingly. We summarize our responses to each comment as follows. We believe that our responses have well addressed all concerns of the reviewer.

*\*\*\* General comments \*\*\**

*\*\*\*\*\*\*\*\*\*\*\*\*\*\*\*\*\*\*\*\*\*\*\*\**

*-> While the new mass-based approach may prevail over area-based calculations, a crucial factor of ozone sensitivity is also the abiotic environment of growth, e.g. water availability, temperature or CO2 concentration. A change in these parameters, all others being equal, may strongly modify the ozone response of plants. It remains unclear how much these confounding factors are already considered in the model by shaping the actual LMA, since the global LMA data are prescribed from data sets. The authors are encouraged to discuss this and, if necessary, amend the model to also consider climatic parameters in their ozone module.*

Response: we have added a new discussion section of 4.4 "Outlook for future modeling" to address this comment (**Lines 398-421**): "In nature, all aspects of plant physiochemical processes, such as growth, development, reproduction, and defense, are influenced by abiotic factors like water

availability, temperature, CO$_2$ concentration, and light resources (Kochhar and Gujral, 2020). In our modeling, the cumulative O$_3$ fluxes are based on dynamic plant simulations with well-established DGVM to calculate the effects of these abiotic factors. LMA is considered as a factor representing the vulnerability of each species, by which divergent responses to the same O$_3$ stomatal dose can be further differentiated. In fact, many other key variables in DGVMs, for example, leaf photosynthetic traits (V$_{cmax}$ and J$_{max}$), nutrient traits (leaf nitrogen and phosphorus), morphological traits (leaf thickness and size), and phenology-related traits (leaf life span) are all more or less interlinked with LMA (Walker et al., 2014). There are some generic regression relationships between them, which have not yet been fully validated by experimental studies. As a result, considerable improvements can be made in the direction of trait-flexible modeling within the existing DGVM frameworks. Our study demonstrates the validity of LMA-based approach for the O$_3$ plant damage modeling.

Although we used the most advanced LMA integrated from available observations, this dataset was developed based on static global grids and revealed the mean state for each pixel. In reality, LMA can vary with biotic/abiotic factors like leaf position in the canopy (Keenan and Niinemets, 2017), phenology, plant health, living environment (Fritz et al., 2018), and climate (Wright et al., 2005; Cui et al., 2020). Even long-term exposure to O$_3$ can alter leaf morphological characteristics and LMA (Li et al., 2017). In future studies, simulations from local to global scales could implement the spatiotemporal variations in LMA taking into account the demographic information and environmental forcings. We expect a breakthrough in the calculation of reliable LMA to achieve fully dynamic predictions of O$_3$ plant damage in Earth System Modeling, thus facilitating the research of plant response and adaption in changing environments."

*-> There are several unclear points in the methods; these are detailed below in the specific comments. I mention them here as they sum up to a general comment.*

Response: we have clarified the relevant expressions for each specific comment.

*-> The calibration partly remains unclear (see below). Most importantly, though, an out-of-sample calibration is missing where each PFT is removed from calibration - for both the unified and the supporting PFT-specific calibration - and the resulting estimate compared particularly for this omitted PFT. This is relevant especially for crops, as they are well apart from the other plant types (e.g. in Figure 2), suggesting that this difference could largely drive calibration and thus the resulting performance be overly optimistic. The perfect fit of S_S to S_O for crops in Figure 6b corroborates this hypothesis.*

Response: we have enclosed an additional test to address this concern as follows and updated the final uncertainty range accordingly. New results are on ***Lines 306-309***: "Finally, we tested a new calibration excluding CRO, the PFT that contributed the most to the calibration biases (shown as orange dashed lines in Fig. S8). The results gave an optimal *a* of 3.2, with global damage of 4.5%. All sensitivity experiments achieved consistent results as the YIBs-LMA simulation with damages ranging from 4.5% to 6.5% and spatial correlation coefficients larger than 0.94."

[Figure]

**Figure S8.** Supplementary calibrations excluding CRO are shown as orange dashed lines. Original calibration in Fig. 3 and 1:1 fitting are shown as dashed pink and light grey, respectively. The new slope, NMB, and r are recalculated and noted in square brackets.

*-> An additional, similar exercise could include another year of ozone data. The current study only uses 2010, for calibration and validation. Another year will have another ozone distribution and thus would be useful to validate the findings.*

Response: We have addressed this comment on **Lines 264-265** as "Notably, such calibration of *a* is robust under different $O_3$ field (see Fig. S2)." Fig. S2 is shown as follows.

[Figure]

**Figure S2.** The calibration and validation with $O_3$ data in the year 2020 from CMIP6 SSP5-8.5 scenario. The forcing data remains the same as YIBs-LMA and calibration procedures are the same as in Fig 3. The new calibration achieved a minor shift of the optimal *a* from 3.5 to 3.6.

*-> All of these new suggestions, once implemented, should then also be considered in the discussion section.*

Response: We have implemented the new discussion section of 4.4 "Outlook for future modeling" (***Lines 398-421***) as suggested.

*** *Specific comments* ***
* * *
*-> Methods, 2.1: if F is the UNdamaged fraction, why is there a ozone penalty in equation 1?*
Response: The damaged fraction should be expressed as $F = a_{PFT} \times max\{f_{O3} - y, 0\}$ .
In our paper, we showed the UNdameged fraction $F = 1 - a_{PFT} \times max\{f_{O3} - y, 0\}$ , which can be directly multiplied on net photosynthetic rate and stomata conductance to calculate the remaining part.

*-> Methods, 2.1: explain f_O3 at first mention and add units to all variables*
Response: we have added related information on ***Lines 149-150*** as "The stomatal $O_3$ flux $f_{O3}$ (nmol m$^{-2}$ s$^{-1}$) is calculated…"

*-> Methods, 2.1, eq 2: f_O3 depends, in turn, on F. Please explain this circular dependence in the text and also what it means for calculation - do you need an optimizing routine?*

Response: we have further explained it on **Lines 165-166** as "Equations (2) and (4) can form a quadratic equation. The *F* can be derived at each timestep (i.e. hourly) and applied to net photosynthetic rate and stomatal conductance to calculate the $O_3$-induced damages."

*-> Methods, 2.1: how do water availability, temperature, CO2 et al. interact with the ozone uptake?*
Response: The ozone uptake is regulated by stomatal conductance, which is dependent on environmental factors such as water availability, temperature, $CO_2$ and so on. In the revised paper, we clarified on **Lines 221-223**: "In this study, all $O_3$ vegetation damage schemes are implemented in the YIBs model (Yue and Unger, 2015), which is a process-based dynamic global vegetation model incorporated with well-established carbon, energy, and water interactive schemes."

We have added the section 4.4 in discussion about abiotic factors. See ***the answers to the first general comment.***

*-> Methods, 2.2, eq 6: what happens with negative values of f_O3 - y in the integral?*
Response: In calculation, y is taken as a threshold, above which the $f_{O3}$ are accumulated. We have modified ***Eqn. 6*** to be $POD_y = \int \max\{f_{O3} - y, 0\}$

*-> Methods, 2.2: is every PFT dominant somewhere?*
Response: Every PFT dominants some regions as ***Figure S1*** shows.

*-> Methods, 2.3: the exact recipe for the calibration is missing. It remains partly elusive how you did the calibration - how many runs, which parameters were tuned, which step size, which algorithm, which target variables etc. Please augment, for all runs.*
Response: We have modified this part on **Lines 211-218** as "For all supporting experiments, the parameter *a* for YIBs-LMA or the eight mean $a_{PFT}$ for YIBs-S2007_adj are derived with the optimal 1:1 fitting between $S_S$ and $S_O$ to minimize the possible biases (Tables 2 and S3-S6). The basic method for calibration is feeding the model with series values of *a* or $a_{PFT}$ until the predicted $O_3$ damage matches observations with the lowest normalized mean biases (NMB). For all LMA-based experiments, $S_S$ from varied PFTs were grouped for the calibration of *a*, while for $a_{PFT}$ in YIBs-S2007_adj, each $a_{PFT}$ is determined individually by matching simulated $S_S$ with $S_O$. Since $S_O$ are available only for six out of the eight YIBs PFTs, including EBF, NF, DBF, $C_3$ grass, $C_4$ grass, and crop (Table S1), $S_O$ of these PFTs are used for calibration. All runs are summarized in Table 1."

*-> Methods, 2.3: a sensitivity towards environmental parameters would be useful to add*
Response: As we explained in the answer to the first general comment, the plant responses to abiotic factors were accounted for in other well-established modules of the vegetation model.

*-> Results, 3.1, l199+: is the higher agreement between observations and mass-based simulations (R2 = 0.77), when compared to area-based simulations (R2 = 0.54), expectable already in the uncalibrated version given the design towards mass-based traits?*
Response: Yes, we reproduced the observed convergence in PFT-level $O_3$ damage in Fig. 2, in which all PFTs showed more consistency in DRRs with LMA-based sensitivity than the area-based approach.

-> *Results, 3.2: can you justify the use of S2007 as a reference, i.e. why is the new model good if it agrees with the old?*

Response: We have added discussion in **Lines 368-378** as: "…The similarity between YIBs-S2007 and YIBs-LMA shown in Fig. 5 revealed an advance in the modeling strategy. Simulated $O_3$ damage in YIBs-S2007 is based on the PFT-level calibrations that tuned sensitivity parameters of each PFT with observed DRRs. Such refinement is a data-driven approach without clear physical reasons. Instead, the YIBs-LMA framework converts the area-based responses to mass-based ones and achieves better unification in $O_3$ sensitivities among different PFTs. In this algorithm, the $O_3$ damage efficiency is inversely related to plant LMA, which influences both the $O_3$ uptake potential and the detoxification capability of the vegetation. The similarity in the global assessment of $O_3$ vegetation damage between YIBs-S2007 and YIBs-LMA further demonstrated the physical validity of LMA-based scheme in the Earth system modeling, because the independent LMA map was applied in the latter approach."

-> *Results, 3.2, l232+: can you provide numbers on the difference components ([O3], LMA variation, land-use intensity etc.)?*

Response: The information of all datasets is shown in section 2.4 **between Lines 223-246** as "…The model applies the same PFT classifications as the Community Land Model (Bonan et al., 2003) (Fig. S1). Eight PFTs are employed including evergreen broadleaf forest (EBF), needleleaf forest (NF), deciduous broadleaf forest (DBF), cold shrub (C_SHR), arid shrubland (A_SHR), $C_3$ grassland (C3_GRA), C4 grassland (C4_GRA), and cropland (CRO) (Fig. S1)…The gridded LMA required for the main mass-based simulation is derived from Moreno-Martinez et al. (2018) (M2018), which shows the highest value of >150 g m$^{-2}$ for needleleaf forest at high latitudes while low values of ~40 g m$^{-2}$ for grassland and cropland (Fig. 1a and Fig. S1). Grids with missing LMA data are filled with the mean of the corresponding PFT. Contemporary $O_3$ concentration fields in the year of 2010 from the multi-model mean in Task Force on Hemispheric Transport of Air Pollutants (TF-HTAP) experiments (Turnock et al., 2018) (Fig. 1b) are used as forcing data. The original monthly $O_3$ data are downscaled to hourly using the diurnal cycle predicted by the chemistry-climate-carbon fully coupled model ModelE2-YIBs (Yue and Unger, 2015). Generally, areas of severe $O_3$ pollution are found in the mid-latitudes of the Northern Hemisphere with highest annual average $O_3$ concentration of over 40 ppbv in East Asia…"

-> *Figure 3h (crops): a linear fit does not seem to be the best choice here, in contrast to all other PFTs. How to account for that or interpret this levelling off?*

Response: Croplands are greatly influenced by human manipulations. To achieve a better simulation, some widely-used DGVMs, like JULES, have their specialized model version for cropland modeling. In the YIBs model, we calculate crop GPP as natural PFTs, but with global map of crop phenology for field regulations (such as plantation and harvest). In future studies, we hope to improve the crop simulations with specific crop modules.

*** *Technical corrections* ***
* * *
*-> Methods, 2.2: explain POD at first mention (abbreviation & what does it mean)*
Response: Corrected on Lines 67-68 as "POD$_y$ (Phytotoxic O$_3$ Dose above a threshold flux of y (Buker et al., 2015))".

*-> Methods, 2.2, l131: what do you mean with 'bio-indicators'?*
Response: We changed the 'bio-indicator' to 'biotic indicator' for clarity.

*-> Results, 3.3: this section requires language proof-reading*
*(Uncertainty section)*
Response: We have proof-read and revised this section as suggested.

*-> Results, 3.3, l250: the values (-0.2 and 1.7) are not %, but percentage points - the difference in %*
*would be much larger*
Response: we have changed the way to describe it on **Lines 308-309** as "All sensitivity experiments achieved consistent results as the YIBs-LMA simulation with damages ranging from 4.5% to 6.5% and spatial correlation coefficients larger than 0.94."

*-> Figure 2: please add the 1:1 line and the out-of-sample line once it is calculated*

[Figure]

Response: we have added 1:1 fitting (light grey lines) for each subplot (see new **Fig. 3**). The recalibration excluding CRO was shown in the new Fig. S8.

*-> Figure 3: add the grey simulated dots to the legend*
Response: we added this legend as suggested. See Fig. 4 as following:

[Figure]

Response: Results for CRO have been added to Fig. 7a in the revised paper:

[Figure]

Reference

Buker, P., Feng, Z., Uddling, J., Briolat, A., Alonso, R., Braun, S., Elvira, S., Gerosa, G., Karlsson, P. E., Le Thiec, D., Marzuoli, R., Mills, G., Oksanen, E., Wieser, G., Wilkinson, M., and Emberson, L. D.: New flux based dose-response relationships for ozone for European forest tree species, Environ Pollut, 206, 163-174, 10.1016/j.envpol.2015.06.033, 2015.

Cui, E., Weng, E., Yan, E., and Xia, J.: Robust leaf trait relationships across species under global environmental changes, Nat Commun, 11, 2999, 10.1038/s41467-020-16839-9, 2020.

Fritz, M. A., Rosa, S., and Sicard, A.: Mechanisms Underlying the Environmentally Induced Plasticity of Leaf Morphology, Front Genet, 9, 478, 10.3389/fgene.2018.00478, 2018.

Keenan, T. F. and Niinemets, U.: Global leaf trait estimates biased due to plasticity in the shade, Nat Plants, 3, ARTN 1620110.1038/nplants.2016.201, 2017.

Kochhar, S. and Gujral, S.: Abiotic and Biotic Stress, in: Plant Physiology: Theory and Applications, 2 ed., edited by: Kochhar, S. L., and Gujral, S. K., Cambridge University Press, Cambridge, 545-589, DOI: 10.1017/9781108486392.021, 2020.

Li, P., Feng, Z., Catalayud, V., Yuan, X., Xu, Y., and Paoletti, E.: A meta-analysis on growth, physiological, and biochemical responses of woody species to ground-level ozone highlights the role of plant functional types, Plant Cell Environ, 40, 2369-2380, 10.1111/pce.13043, 2017.

Moreno-Martinez, A., Camps-Valls, G., Kattge, J., Robinson, N., Reichstein, M., van Bodegom, P., Kramer, K., Cornelissen, J. H. C., Reich, P., Bahn, M., Niinemets, U., Penuelas, J., Craine, J. M., Cerabolini, B. E. L., Minden, V., Laughlin, D. C., Sack, L., Allred, B., Baraloto, C., Byun, C., Soudzilovskaia, N. A., and Running, S. W.: A methodology to derive global maps of leaf traits using remote sensing and climate data, Remote Sens Environ, 218, 69-88, 2018.

Turnock, S. T., Wild, O., Dentener, F. J., Davila, Y., Emmons, L. K., Flemming, J., Folberth, G. A., Henze, D. K., Jonson, J. E., Keating, T. J., Kengo, S., Lin, M., Lund, M., Tilmes, S., and O'Connor, F. M.: The impact of future emission policies on tropospheric ozone using a parameterised approach, Atmos Chem Phys, 18, 8953-8978, 10.5194/acp-18-8953-2018, 2018.

Walker, A. P., Beckerman, A. P., Gu, L., Kattge, J., Cernusak, L. A., Domingues, T. F., Scales, J. C., Wohlfahrt, G., Wullschleger, S. D., and Woodward, F. I.: The relationship of leaf photosynthetic traits – Vcmax and Jmax – to leaf nitrogen, leaf phosphorus, and specific leaf area: a meta-analysis and modeling study, Ecol Evol, 4, 3218-3235, https://doi.org/10.1002/ece3.1173, 2014.

Wright, I. J., Reich, P. B., Cornelissen, J. H. C., Falster, D. S., Groom, P. K., Hikosaka, K., Lee, W., Lusk, C. H., Niinemets, U., Oleksyn, J., Osada, N., Poorter, H., Warton, D. I., and Westoby, M.: Modulation of leaf economic traits and trait relationships by climate, Global Ecol Biogeogr, 14, 411-421, 10.1111/j.1466-822x.2005.00172.x, 2005.

Yue, X. and Unger, N.: The Yale Interactive terrestrial Biosphere model version 1.0: description, evaluation and implementation into NASA GISS ModelE2, Geosci Model Dev, 8, 2399-2417, 10.5194/gmd-8-2399-2015, 2015.

---

## Author Comment (AC2)

**Response to the review comment 2 on gmd-2022-227**

*Review for GMD-2022-227 by Yimian Ma et al.*

*-------------------------------------------*

*This paper addresses the importance issue of properly representing interspecific variations of plant sensitivity to ozone damage in global ecosystem or Earth system models, by taking advantage of the observed relationships between leaf-based traits (such as leaf mass per area) and ozone sensitivity. The methodology and analysis are scientifically rigorous and valid, and potentially important implications for all future studies of plant-ozone interactions. I recommend the publication of this manuscript as long as the following suggestions have been addressed.*

Response: Thank you very much for reviewing this manuscript and offering constructive suggestions for further improvement. We have considered them carefully and revised our manuscript accordingly, especially for the Introduction and Discussion parts. In this version, we have added more explanations and comparisons of previous modeling schemes and fully demonstrated the necessity of this study. Here are point-to-point responses:

*Section 1:*

*Overall, the introduction is too short, thus the motivation and justification for the importance of their work are relatively weak. It is also not as informative as what an introduction section should be like. The authors are thus recommended to lengthen the introduction, especially to:*

*How exactly are the different kinds of plant sensitivities currently used in models measured/ determined? What are the differences between the different approaches (e.g., Felzer vs. Lombardozzi vs. Sitch)? Based on experimental values or field observations? A discussion on the methodological and theoretical basis of the current approaches should be included. Moreover, a comparative analysis of the numerical results from the different approaches and studies should be included to highlight the uncertainties and justify the need to revise the current approach.*

Response: In the revised Introduction, we explained the theoretical basis of different schemes and compared their numerical results for the global estimates on ***Lines 84-97*** as follows: "Alternatively, more and more mechanistic schemes were developed and implemented in dynamic global vegetation models (DGVMs) to assess the joint effects of environmental factors and $O_3$ on plants. Felzer et al. (2004) considered both the damaging (through AOT40) and healing (through growth) processes related to $O_3$ effects within the framework of Terrestrial Ecosystem Model. They further estimated the reduction of 2.6%-6.8% in the net primary productivity by $O_3$ pollution in U.S. during 1980-1990. Different from Felzer et al. (2004), Sitch et al. (2007) proposed a flux-based scheme linking the instantaneous $POD_y$ with the damaging percentage through the coupling between stomatal conductance and photosynthetic rate. Implementing this scheme into the vegetation model of YIBs, Yue and Unger (2015) predicted a range of 2%-5% reduction in global gross primary productivity (GPP) taking into account the low to high $O_3$ sensitivities for each vegetation types.

Lombardozzi et al. (2015) collected hundreds of measurements and derived the decoupled responses in stomatal conductance and photosynthesis for the same $O_3$ uptake fluxes. They further implemented the separate response relationships into the Community Land Model and estimated a reduction of 8%-12% in GPP by $O_3$ at present day."

The $O_3$ sensitivities in these schemes are further explained on *lines 101-105* as follows: "Although different schemes considered varied physical processes (Ollinger et al., 1997; Felzer et al., 2004; Sitch et al., 2007; Felzer et al., 2009; Lombardozzi et al., 2015; Oliver et al., 2018), they followed the same principle that different $O_3$ sensitivities should be applied for varied plant functional types (PFTs), as revealed by many measurements in the past four decades (Buker et al., 2015; Mills et al., 2018) (Table S1)."

*In addition to semi-mechanistic representation of sensitivity of photosynthesis to ozone exposure, there have also been other more empirical approaches to quantify plant sensitivity to ozone, including the concentration-based approach (e.g., AOT40) and flux-based approach (e.g., $DO_3SE$, POD). These approaches have been mostly applied to crops but also to some extent to natural vegetation. A paragraph should be devoted to discuss the merits and shortfalls of these various approaches, so as to justify the importance of mechanistically representing photosynthetic responses to ozone exposure. Some references that should be discussed include Tai et al. (2021) and Emberson et al. (2018).*

Response: We added the introductions of concentration- and flux-based metrics on *Lines 63-71* as: "To date, $O_3$ fumigation experiments have been conducted for various plant species. Accordingly, $O_3$ damaging sensitivities, denoted as the Dose-Response Relationships (DRRs), were derived as the regressions between $O_3$ exposure metrics and the changes in biotic indicators (Mills et al., 2011). The widely-used $O_3$ metrics include ambient $O_3$ concentrations for AOT40 (Accumulated $O_3$ concertation above the Threshold of 40 ppbv (Fuhrer et al., 1997)), or the stomatal $O_3$ flux for $POD_y$ (Phytotoxic $O_3$ Dose above a threshold flux of y (Buker et al., 2015)). The biotic indicators include visual leaf states, photosynthetic rate, biomass, or crop yield. Normally, the DRRs were derived for typical tree/grass species at specific regions, for example, Norway spruce, birch, and beech in Europe (Buker et al., 2015) or poplar (Shang et al., 2017) and crops (Peng et al., 2019) in East Asia."

We added corresponding paragraphs on *Lines 73-82* as "Some assessment studies used DRRs to derive contemporary $O_3$ plant damage patterns at the large scales. Concentration-based DRRs were widely measured and applied on the homogenized land cover, mostly for estimating crop yield loss (Feng et al., 2022; Tai et al., 2021; Hong et al., 2020). However, such DRRs don't include information about biochemical defense and stomatal regulations. Comparatively, flux-based DRRs reflect a more reasonable consideration in biological processes, but are limited by the application scales in both space and time (Mills et al., 2011; Mills et al., 2018). For example, the estimate of $POD_y$ needs a dry deposition model "$DO_3SE$" (Deposition of Ozone for Stomatal Exchange) (Clrtap, 2017) or an equivalent model to account for environmental constraints on plant stomatal uptake during the whole growing season. Furthermore, the application of DRRs might introduce uncertainties due to the omission of complex interactions among biotic and abiotic factors at varied spatiotemporal scales."

*A proper representation of ozone-vegetation interactions is important in Earth system and atmospheric modeling as much as in ecosystem modeling, because ozone damage on plants can subsequently affects land surface fluxes and thus atmospheric chemistry and climate. Some discussion should be done on these aspects, with references to, e.g., Zhou et al. (2018), Gong et al. (2020), and Zhu et al. (2022).*

Response: we have added a paragraph to describe such importance, see ***Lines 97-99:*** "Coupling these schemes with earth system models, studies have assessed interactive $O_3$ impacts on carbon sink (Oliver et al., 2018; Yue and Unger, 2018), global warming (Sitch et al., 2007), and air pollution (Zhou et al., 2018; Gong et al., 2020; Gong et al., 2021; Zhu et al., 2022)."

*The possible theoretical basis behind the connection between LMA and ozone sensitivity should be discussed. Possible uses of equations are recommended.*

Response: the possible theoretical basis is summarized on ***Lines 319-321*** as "Moreover, it seems plausible that the oxidative stress caused by a given amount of stomatal $O_3$ flux per unit leaf area would be distributed over a larger leaf mass, and hence diluted, in a leaf with high LMA." We have added brief introduction on ***Lines 122-123*** as "This is likely related to the diluting effect of thicker leaves, which normally have stronger defenses against $O_3$ in their cross-section." An equation for summarizing such a phenomenon is ***Eqn. 3***, which reveals an inverse relationship between plant $O_3$ sensitivity and LMA. ***Figs 8 and 9*** also show this message.

*Section 2.2:*

*The authors describe the POD approach here. As mentioned above, a due discussion comparing the various approaches including POD should be given in the introduction section.*

Response: as have answered before, a brief introduction of POD has been added ***on Lines 67-69*** as one of the flux-based metrics. Besides, its physiochemical meaning, calculation, and limitation in the direct application have been further demonstrated between ***Lines 76-82***.

*Section 2.3:*

*Since the calibration exercise is so crucial to this study, the authors are recommended to include at least a table or two for the calibration (from Table S3–S7), ideally the most important one or a consolidated one, in the main paper. Details of the calibration method (e.g., Monte Carlo? Or simply varying the value manually until it fits the best?) should be given in the text or table caption.*

Response: The original Table S3 has been moved into the main text as ***Table 2***. A more thorough description of calibration has been added in Method in ***Lines 211-218*** as "For all supporting experiments, the parameter $a$ for YIBs-LMA or the eight mean $a_{PFT}$ for YIBs-S2007_adj are derived with the optimal 1:1 fitting between $S_S$ and $S_O$ to minimize the possible biases (Tables 2 and S3-S6). The basic method for calibration is feeding the model with series values of $a$ or $a_{PFT}$ until the

predicted $O_3$ damage matches observations with the lowest normalized mean biases (NMB). For all LMA-based experiments, $S_S$ from varied PFTs were grouped for the calibration of $a$, while for $a_{PFT}$ in YIBs-S2007_adj, each $a_{PFT}$ is determined individually by matching simulated $S_S$ with $S_O$. Since $S_O$ are available only for six out of the eight YIBs PFTs, including EBF, NF, DBF, $C_3$ grass, $C_4$ grass, and crop (Table S1), $S_O$ of these PFTs are used for calibration. All runs are summarized in Table 1."

*Section 2.4:*

*Since the global distribution of ozone concentration is so crucial in evaluating the resulting GPP reductions, the global map of ozone concentration should be given in the main text instead of in the supplementary materials.*

Response: We have moved the original Fig. S1 to ***Fig.1*** in the main context.

*Section 3:*

*The use of tenses seems to be inconsistent across the paper. Section 2 mostly uses the present tense, but the past tense is sometimes used in Section 3. The authors are recommended to consistently use tenses throughout the paper (i.e., using the past tense for the research tasks and actions they did for their study and for the actions done by previous researchers, but the present tense whenever the results are presented and discussed).*

Response: We have checked and revised tenses accordingly. Please see the manuscript with tracking changes.

*Section 3.2:*

*It is not surprising that "the simulation with the optimal a =3.5 nmol-1 s g predicted a global GPP reduction of 4.8% (Fig. 4a), which was similar to the value estimated with the area-based S2007 scheme", because ultimately the LMA approach is derived from the area-based approach. This then comes to an important question – why do we need to use the LMA approach after all, if the resulting GPP is similar? This should be addressed. I suspect that using the LMA approach may better capture the regional differences and intra-PFT variations, but these are not explicitly shown or analyzed by the authors, who are thus recommended to address these issues (e.g., by elaborately comparing the PFT-specific and/or regional differences of ozone damage from the area-based approach vs. the LMA approach). This is done in part in Fig. 4, but the attribution to PFT or regional variations are lacking. It may be important to show how each PFT behaves differently under the two approaches.*

Response: We added some discussion in ***4.3*** on ***Lines 368-378*** to explain why it is an advance in the modeling strategy as follows: "…The similarity between YIBs-S2007 and YIBs-LMA shown in Fig. 5 revealed an advance in the modeling strategy. Simulated $O_3$ damage in YIBs-S2007 is based on the PFT-level calibrations that tuned sensitivity parameters of each PFT with observed DRRs. Such refinement is a data-driven approach without clear physical reasons. Instead, the YIBs-LMA framework converts the area-based responses to mass-based ones and achieves better unification in

O₃ sensitivities among different PFTs. In this algorithm, the O₃ damage efficiency is inversely related to plant LMA, which influences both the O₃ uptake potential and the detoxification capability of the vegetation. The similarity in the global assessment of O₃ vegetation damage between YIBs-S2007 and YIBs-LMA further demonstrated the physical validity of LMA-based scheme in the Earth system modeling, because the independent LMA map was applied in the latter approach."

For other advantages, we have summarized in the previous version, which is now in *4.3* on *Lines 367-395* as: "For the first time, we implemented plant trait LMA into a process-based O₃ impact modeling scheme and obtained reasonable interspecific and inter-PFT O₃ responses supported by observations.… In addition to the advance in physical mechanisms, the LMA-based approach improves global O₃ damage assessments in the following aspects. First, it significantly reduces the number of required key parameters. To account for interspecific sensitivities, many schemes have to define PFT-level parameters to cap the ranges of plant responses (Sitch et al., 2007; Felzer et al., 2009; Lombardozzi et al., 2015). As a result, those schemes rely on dozens of parameters which increases the uncertainties of modeling and the difficulties for model calibration. The LMA-based approach requires the calibration of one single parameter *a*, largely facilitating its application across different vegetation models. Second, the new approach accounts for the continuous spectrum of O₃ sensitivities. Previous studies usually categorized species into groups of low or high O₃ sensitivity, depending on very limited data from O₃ exposure experiments. As a result, gridcells for a specific PFT share the same sensitivities regardless of their geographic locations and ecosystem characteristics. In reality, there are hundreds and thousands of plant species in each PFT and they usually have large variations in biophysical parameters including LMA and O₃ sensitivities. The LMA-based approach takes advantage of the newly revealed unifying concept in O₃ sensitivity (Li et al., 2016; Feng et al., 2018; Li et al., 2022) and the recent development in a trait-based LMA global map (Fig. 1a). Such configurations present a spectrum of gridded O₃ sensitivities (Fig. 8a) following the variations of LMA distribution."

In *Table S8*, PFT-specific comparisons for different simulations are shown as detailed data. We further plot a picture below. Generally, two schemes show comparative simulation capacity in our global simulation from a PFT-based statistic perspective.

[Figure]

**Figure S5.** Comparison of PFT-specific GPPs from YIBs-LMA and YIBs-S2007_adj. Data for each PFT are shown as bars with blue, red, and green representing different experiments. Ratio numbers above each group of bars reveal the PFT-specific damage ratios for simulations using two schemes with red and green representing YIBs-LMA and YIBs-S2007_adj, respectively.

*A more elaborate discussion should be given to how "the differences in LMA and simulated O3 sensitivities of these PFTs were the main cause of discrepancies in GPP damage at the large scale".*

Response: this sentence has been modified to "However, the differences in LMA and simulated $O_3$ sensitivities of these PFTs also made important contributions to such discrepancies in GPP damages." on **Lines 290-291**.

*Section 4:*

*The authors have described the possible mechanisms behind the LMA-ozone damage relationships here. As suggested above, some of these should be devoted to the introduction section (at least discussed briefly), and here the authors may discuss how their model development and simulations verify them and allow them to derive a fuller picture.*

Response: we have added a brief sentence on **Lines 122-123** as "This is likely related to the diluting effect of thicker leaves, which normally have stronger defenses against $O_3$ in their cross-section." In the discussion part, we explained how we interpreted this idea in our algorithm "Simulated $O_3$ damage in YIBs-S2007 is based on the PFT-level calibrations that tuned sensitivity parameters of each PFT with observed DRRs. Such refinement is a data-driven approach without clear physical reasons. Instead, the YIBs-LMA framework converts the area-based responses to mass-based ones and achieves better unification in $O_3$ sensitivities among different PFTs. In this algorithm, the $O_3$ damage efficiency is inversely related to plant LMA, which influences both the $O_3$ uptake potential and the detoxification capability of the vegetation." (**Lines 370-375**)

*Section 4.3:*

*The authors well justify the merits of their LMA-based approach. Indeed, this can bring potentially significant unification and simplification of global modeling. I would further recommend an additional merit is that the LMA-based approach can even address the intra-PFT (not just inter-PFT) variations in ozone sensitivity because species in the same PFT can have largely varying LMA. Even though for now each PFT may have a fixed LMA in many models, this LMA-based approach provides a unifying way to model ozone damage as more spatially resolved LMA data become available in the future.*

Response: We have strengthened the advantages of the LMA-based approach in **the Introduction part in Lines 73-115** and the **Discussion section 4.3**. Besides, we have also added a **Section 4.4**

***outlook for future modeling*** to further discuss the prospect of the LMA-based approach "In nature, all aspects of plant physiochemical processes, such as growth, development, reproduction, and defense, are influenced by abiotic factors like water availability, temperature, $CO_2$ concentration, and light resources (Kochhar and Gujral, 2020). In our modeling, the cumulative $O_3$ fluxes are based on dynamic plant simulations with well-established DGVM to calculate the effects of these abiotic factors. LMA is considered as a factor representing the vulnerability of each species, by which divergent responses to the same $O_3$ stomatal dose can be further differentiated. In fact, many other key variables in DGVMs, for example, leaf photosynthetic traits ($V_{cmax}$ and $J_{max}$), nutrient traits (leaf nitrogen and phosphorus), morphological traits (leaf thickness and size), and phenology-related traits (leaf life span) are all more or less interlinked with LMA (Walker et al., 2014). There are some generic regression relationships between them, which have not yet been fully validated by experimental studies. As a result, considerable improvements can be made in the direction of trait-flexible modeling within the existing DGVM frameworks. Our study demonstrates the validity of LMA-based approach for the $O_3$ plant damage modeling.

Although we used the most advanced LMA integrated from available observations, this dataset was developed based on static global grids and revealed the mean state for each pixel. In reality, LMA can vary with biotic/abiotic factors like leaf position in the canopy (Keenan and Niinemets, 2017), phenology, plant health, living environment (Fritz et al., 2018), and climate (Wright et al., 2005; Cui et al., 2020). Even long-term exposure to $O_3$ can alter leaf morphological characteristics and LMA (Li et al., 2017). In future studies, simulations from local to global scales could implement the spatiotemporal variations in LMA taking into account the demographic information and environmental forcings. We expect a breakthrough in the calculation of reliable LMA to achieve fully dynamic predictions of $O_3$ plant damage in Earth System Modeling, thus facilitating the research of plant response and adaption in changing environments."

*References to cite:*

- *Emberson, L. D., Pleijel, H., Ainsworth, E. A., van den Berg, M., Ren, W., Osborne, S., ..., & Van Dingenen, R. (2018). Ozone effects on crops and consideration in crop models. European Journal of Agronomy, 100, 19-34.*

- *Gong, C., Lei, Y., Ma, Y., Yue, X., & Liao, H. (2020). Ozone–vegetation feedback through dry deposition and isoprene emissions in a global chemistry–carbon–climate model. Atmospheric Chemistry and Physics, 20(6), 3841–3857.*

- *Tai, A. P. K., Sadiq, M., Pang, J. Y. S., Yung, D. H. Y., & Feng, Z. (2021). Impacts of surface ozone pollution on global crop yields: Comparing different ozone exposure metrics and incorporating co-effects of $CO_2$. Frontiers in Sustainable Food Systems, 5, 534616.*

- *Zhou, S. S., Tai, A. P. K., Sun, S., Sadiq, M., Heald, C. L., & Geddes, J. A. (2018). Coupling between surface ozone and leaf area index in a chemical transport model: strength of feedback and implications for ozone air quality and vegetation health. Atmospheric Chemistry and Physics, 18(19), 14133-14148.*

- *Zhu, J., Tai, A. P. K., & Yim, S. H. L. (2022). Effects of ozone-vegetation interactions on meteorology and air quality in China using a two-way coupled land-atmosphere model. Atmospheric Chemistry and Physics, 22(2), 765-782.*

Response: All these references have been cited.

Reference

[revised manuscript text omitted]

---

## Author Response (AR2)

**Response to the reviewer #1 on gmd-2022-227**

*Review for GMD-2022-227 by Yimian Ma et al.*
* * *
Response: Thank you for your time and effort in reviewing our paper gmd-2022-227, and for agreeing to its publication. We appreciate all your valuable comments and feedback during the whole process, which have been immensely helpful in improving the quality of our work.

**Response to the reviewer #2 on gmd-2022-227**

*Review for GMD-2022-227 by Yimian Ma et al.*
* * *
*The revised manuscript has addressed the scientific concerns I have raised adequately and can be published. Some final minor technical corrections are recommended to correct new grammar and writing errors that appear in the revised text (e.g., inappropriate use of short forms "don't" in L73-82, inconsistent use of past vs. present tense, etc.).*

Response: Thank you very much for taking the time to provide additional comments and feedback on our paper GMD-2022-227. We have carefully rechecked our manuscript and have made the necessary changes to address the final minor technical corrections for grammar and writing errors. Due to there is no option for uploading tracked file in system for this round of revision, please find minor revisions in the new version of manuscript and sorry for the inconvenience. We believe that these changes have further improved the manuscript and have addressed any remaining concerns that you may have had. We appreciate the time and effort you have invested in reviewing our paper, which is immensely beneficial in improving our work. Thank you very much for agreeing to its publication.